# Probabilistic Emulation of a Global Climate Model with Spherical DYffusion

**Salva Rühling Cachay**
UC San Diego

**Brian Henn**
Allen Institute for AI

**Oliver Watt-Meyer**
Allen Institute for AI

**Christopher S. Bretherton**
Allen Institute for AI

**Rose Yu**
UC San Diego

## Abstract

Data-driven deep learning models are transforming global weather forecasting. It is an open question if this success can extend to climate modeling, where the complexity of the data and long inference rollouts pose significant challenges. Here, we present the first conditional generative model that produces accurate and physically consistent global climate ensemble simulations by emulating a coarse version of the United States' primary operational global forecast model, FV3GFS. Our model integrates the dynamics-informed diffusion framework (DYffusion) with the Spherical Fourier Neural Operator (SFNO) architecture, enabling stable 100-year simulations at 6-hourly timesteps while maintaining low computational overhead compared to single-step deterministic baselines. The model achieves near gold-standard performance for climate model emulation, outperforming existing approaches and demonstrating promising ensemble skill. This work represents a significant advance towards efficient, data-driven climate simulations that can enhance our understanding of the climate system and inform adaptation strategies.[1]

## 1 Introduction

Climate models are foundational tools used to understand how the Earth system evolves over long time periods and how it may change as a response to possible greenhouse gas emission scenarios. Such climate simulations are currently very expensive to generate due to the computational complexity of the underlying physics-based climate models, which must be run on supercomputers. As a result, scientists and policymakers are limited to exploring only a small subset of possibilities for different mitigation and adaptation strategies [48].

Training relatively cheap-to-run data-driven surrogates to emulate global climate models could provide a compelling alternative [15]. Although recent deep learning models are on the verge of transforming the conceptually similar field of medium-range weather forecasting [5, 38, 11, 51], these advances do not directly transfer to long-term climate projections [37]. Indeed, most such models only report forecasts up to two weeks into the future and may diverge or become physically inconsistent over longer simulations. In contrast, climate projections demand accurate and stable simulations of the global Earth system spanning decades or centuries, requiring reliable reproduction of long-term statistics.

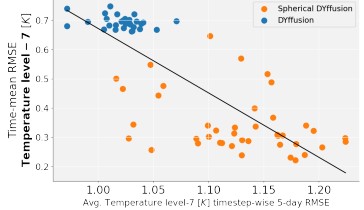

Figure 1: Weather performance (x-axis) is not a strong indicator of climate performance (y-axis). Each dot corresponds to a distinct sample or checkpoint epoch.

---

[1]Code is available at https://github.com/Rose-STL-Lab/spherical-dyffusion

38th Conference on Neural Information Processing Systems (NeurIPS 2024).

In Figure 1 we quantitatively show this divergence between the medium-range weather forecasting skill of ML models (measured as the average RMSE on 5-day forecasts) and their performance on longer climate time scales (measured as the RMSE of the 10-year time-mean). We have verified that this finding holds regardless of the analyzed variable and the proxy used for weather performance, which we discuss in more detail in Appendix E.3. Heuristically, optimizing weather skill ensures that a climate model takes a locally accurate path around the climate 'attractor', but it does not guarantee that small but systematic errors may not build up to distort that simulated attractor to have biased long-term climate statistics. While this is a little-discussed observation in the ML community, the climate modeling community has documented it for physics-based models [17, 54].

A recent breakthrough is a deterministic surrogate called ACE (Ai2 Climate Emulator) [67], which remains remarkably stable and physically consistent over 10-year simulations at 6-hourly time steps, forced by time-varying specified sea-surface temperature and sea-ice. Its success can be attributed to careful data processing, problem design, and the Spherical Fourier Neural Operator (SFNO) [8] architecture. ACE is trained to emulate the United States' primary operational global forecast model, the physics-based FV3GFS [73], which is operationally used at the US National Weather Service and US National Centers for Environmental Prediction. ACE produces encouragingly small ten-year mean climate biases (i.e. biased long-term averages), but they are still significantly larger than the theoretical minimum imposed by internal variability of the reference physics-based model.

ACE's deterministic nature restricts its ability to model the full distribution of climate states or to facilitate ensemble simulations, which involve drawing multiple samples from the same model. These capabilities are crucial for climate modeling, as they enable better uncertainty quantification, more robust and physically consistent predictions, and a deeper understanding of potential future climate scenarios and associated risks [32]. While it is possible to ensemble a deterministic model by perturbing its inputs, this approach often leads to under-dispersed (i.e. overly confident) ensembles compared to generative or physics-based approaches [57]. Even then, the problem remains that due to optimizing them on MSE-based loss functions, the deterministic predictions may degrade to a mean prediction for longer forecast time scales and underestimate unlikely events [9].

A generative modeling approach, particularly the use of diffusion models [59, 25], appears to be a promising solution to these challenges. However, standard diffusion models are computationally intensive to train and sample from. This complexity poses significant problems for climate modeling because: 1) atmospheric data is extremely high-dimensional, making the use of video diffusion models [63, 27, 69, 58, 26, 23] prohibitive, even more so as this class of models still struggle with videos longer than a few seconds; and 2) the sampling speed of standard diffusion models is particularly problematic for long, sequential inference rollouts. For instance, generating a single 10-year-long simulation, as in our experiments, with a standard autoregressive diffusion model [35, 51] that uses $N$ diffusion steps would require $14600 \times N$ neural network forward passes. If a second-order solver for sampling is used [31, 51], this number doubles. Even with $N$ as small as 30, this results in half a million forward passes to generate a single sample trajectory, severely limiting the potential of data-driven models to serve as fast surrogates for expensive physics-based models.

As a solution to this computational problem, we build upon the dynamics-informed diffusion model framework, DYffusion, from Rühling Cachay et al. [57], which caps the computational overhead at inference time (as measured by the number of neural net forward passes) to less than $3 \times$ as much as for a deterministic next-step forecasting model such as SFNO or ACE. Unfortunately, the original DYffusion method relies on an UNet-based architecture designed for Euclidean data rather than physical fields on a sphere. As we show in Figure 1, this mismatch of inductive biases becomes more problematic at the long climate time scales that we focus on in this paper.

We address these limitations by carefully integrating the DYffusion framework with the SFNO architecture from Bonev et al. [8], and the data and evaluation procedure from Watt-Meyer et al. [67]. To achieve this integration, we extend SFNO with time conditioning and inference stochasticity modules. Our proposed framework, Spherical DYffusion, achieves strong results: On average, across all 34 predicted fields, our model reduces climate biases to within $50\%$ of the reference model, which is more than $2 \times$ and $4 \times$ lower than the best baselines. For critical fields, such as the derived total water path quantity, our method achieves results within $20\%$ of the reference model, representing a $5 \times$ improvement over the next best baseline (see Fig. 2). Additionally, our method proves effective for ensemble climate simulations, reproducing climate variability consistent with the reference model and further reducing climate biases towards the theoretical minimum through ensemble-averaging.

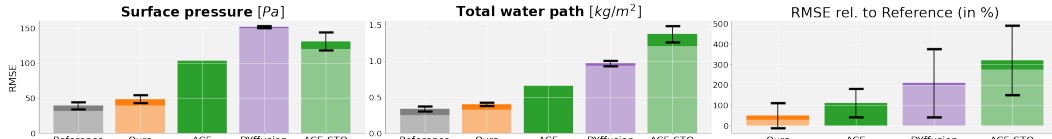

Figure 2: RMSE of 10-year time-means for a subset of important fields. The leftmost bar in the first two subplots shows the reference noise floor, determined by comparing ten independent 10-year reference FV3GFS simulations with the validation simulation. The scores computed using the mean over these ten simulations (a proxy for an "ensemble prediction") are shown in light shade. The subsequent bars show the corresponding scores for our method and the deep-learning baselines, using a 25-member ensemble for the probabilistic methods (all except ACE, which only reports scores for its single deterministic prediction). Scores computed using the ensemble-mean prediction are shown in light shade. The dark shaded bar on top indicates the performance drop when using a single member's prediction only, with error bars representing the standard deviation over the 25 different member choices. The rightmost subplot displays the average time-mean RMSE of the ML-based emulators relative to the reference across all 34 variables. On average, our method's time-mean RMSEs are 50% higher than the noise floor, which is less than half the average RMSE of the next best method, ACE. When using the 25-member ensemble mean prediction, this reduces to 29.28%.

Our generative model is a leap forward toward purely ML-based large ensemble climate projections that are both efficient and accurate. Our main contributions are:

1. We present the first conditional generative model for probabilistic emulation of a realistic climate model, with minimal computational overhead over deterministic baselines.

2. We carefully integrate two distinct frameworks, ACE and DYffusion, including additional modifications to the SFNO architecture such as time-conditioning modules.

3. We show that our integrated method performs considerably better than relevant baselines in terms of reduced climate biases, ensemble-based climate modeling, and consistent variability of the climate predictions.

4. We show that short-term weather performance does not necessarily translate to accurate reproduction of long-term climate statistics.

## 2   Related Work

**ML for weather and climate modeling.**     There are fundamental differences in weather and climate modeling. Climate refers to the average weather over long periods of time[2]. While weather forecasting focuses on short time scales in the order of days or weeks, climate modeling simulates longer periods of decades to centuries. Weather forecasting is primarily an initial-value problem, for which it is important to analyze short-term time-specific predictions. Climate modeling is primarily a boundary-condition (or forcing-driven) problem [65], characterized by long-term averages and distributions.

Deep learning-based models have emerged as a much more computationally efficient alternative to traditional physics-based numerical weather prediction (NWP) models, showing impressive skill for deterministic medium-range weather forecasting [49, 33, 5, 8, 10, 47, 38]. This success has been more recently extended to ensemble-based probabilistic weather forecasting [34, 51]. An alternative approach is hybrid modeling, where a physics-based component is complemented by ML-based parameterizations or corrections [52, 71, 56, 1, 36, 70, 34]. At longer lead times, when weather becomes chaotic and less predictable, the ensemble mean prediction of a physics-based or probabilistic ML-based ensemble improves deterministic metrics such as root mean squared error (RMSE) over non-ensembled methods [51, 34, 53].

However, advances in weather forecasting hardly transfer to long-term climate projections. Fully data-driven models fail to maintain stability beyond two-week-ahead forecasts, as errors accumulate over their autoregressive rollouts. Weyn et al. [68] and Bonev et al. [8] showed stable forecasts for horizons of up to six weeks and one year, respectively. Only recently, Watt-Meyer et al. [67] notably achieved stable and accurate 10-year simulations, followed by another deterministic SFNO-based climate emulator showing promising results using four prognostic variables [22]. Easier, but less flexible and

---

[2]For example, see https://oceanservice.noaa.gov/facts/weather_climate

informative, alternatives to full-scale temporal modeling of atmospheric dynamics, include emulation of annual means given an emission scenario [66, 30, 46, 42], temporal super-resolution of monthly means [4], or debiasing climate model output [3, 7, 45].

**Diffusion models.** Diffusion models [25, 59–61] have demonstrated significant success in generating data such as natural images and videos. While traditionally formulated for finite-dimensional spaces, these models have been extended to function spaces [40]. Their direct applications to autoregressive forecasting [35, 51] and downscaling [64, 43, 20] of physical data have shown promising results. However, these approaches inherit the computational complexity associated with training and sampling from standard diffusion models. This is particularly prohibitive for autoregressive predictions on climate time scales, as the total number of neural network forward passes increases proportionally with the number of sampling steps, typically ranging from 20 to 1000. Consequently, recent research that leverages insights from diffusion models to balance predictive performance and sampling speed appears more promising for assessing their viability in climate simulations [57, 41]. While diffusion models traditionally rely on U-Net architectures [55, 13], vision transformers have shown promising results in image synthesis [50, 29, 24]. Our work explores a different, neural operator-based, architecture for Earth data.

## 3 Background

We first define the problem and then introduce the key components in our framework, namely DYffusion and SFNO. We abbreviate a time series of tensors $\boldsymbol{y}_0, \dots, \boldsymbol{y}_t$ with $\boldsymbol{y}_{0:t}$.

### 3.1 Problem Setting

Our goal is to learn the probability distribution $P(\boldsymbol{x}_{1:H} \mid \boldsymbol{x}_0, \boldsymbol{f}_{0:H})$ over a horizon of $H$ time steps, conditional on initial conditions $\boldsymbol{x}_0$ and a scenario of forcing variables $\boldsymbol{f}_{0:H}$ (i.e. time-varying boundary conditions). In our paper, these forcings correspond to prescribed sea surface temperatures and incoming solar radiation (see Section 5.1), leaving it to future work to force based on greenhouse gas emission scenarios explicitly. Each $\boldsymbol{x}_t \in \mathbb{R}^D$ represents the state of the atmosphere at a given timestep, $t$, consisting of two- and three-dimensional surface and atmospheric variables across a latitude-longitude grid. These variables, which serve as both input and output, are referred to as *prognostic* variables. We assume a constant time interval between successive time steps $t$ and $t+1$. To make training feasible, it is necessary to train on a much shorter horizon $h$, i.e. learn the distribution $P(\boldsymbol{x}_{t+1:t+h} \mid \boldsymbol{x}_t, \boldsymbol{f}_{t:t+h})$, and apply the model autoregressively. This process begins with $P(\boldsymbol{x}_{1:h} \mid \boldsymbol{x}_0, \boldsymbol{f}_{0:h})$ and continues until reaching time step $H$ at inference time.

### 3.2 Diffusion Models and DYffusion

Diffusion models can be seen as a general paradigm to learn the target distribution $p(\mathbf{s}^{(0)})$, by iterating over $N$ diffusion steps of a forward or reverse process. We denote the states of each diffusion step with $\mathbf{s}^{(n)}$, using a superscript $n$ to clearly distinguish them from the physical time steps of the data $\boldsymbol{x}_t$. Standard diffusion models [59, 25, 31], initialize the reverse process from a simple isotropic Gaussian distribution $\mathbf{s}^{(N)} \sim \mathcal{N}(\mathbf{0}, \mathbf{I})$ so that as $n \to 0$ the intermediate states $\mathbf{s}^{(n)}$ are gradually denoised towards a real data sample $\mathbf{s}^{(0)}$.

In Cold Diffusion [2], this paradigm is extended to more general data corruption processes such as blurring. Rühling Cachay et al. [57] propose DYffusion, by adapting cold diffusion models to forecasting problems. The key idea is to make the forward and reverse processes dynamics-informed by directly coupling them to the physical time steps of the data. That is, the reverse process is initialized with $\mathbf{s}^{(N)} = \boldsymbol{x}_0$ and iteratively evolves jointly with the dynamics of the data $\boldsymbol{x}_1, \dots, \boldsymbol{x}_{h-1}$ to reach the data at some target time step, $\mathbf{s}^{(0)} = \boldsymbol{x}_h$.

In DYffusion, the forward and reverse processes are informed by temporal dynamics in the data and do not rely on data corruption. Their only source of stochasticity comes from using a stochastic neural network as an operator for the forward process and is implemented by using Monte Carlo (MC) dropout [19]. This forward process essentially corresponds to a temporal interpolator network, while the reverse process is represented by a multi-step forecasting network. Thus, compared to standard diffusion models, DYffusion requires training one more neural network, which they propose doing in separate stages, beginning with the interpolator model. Due to its dynamics-informed

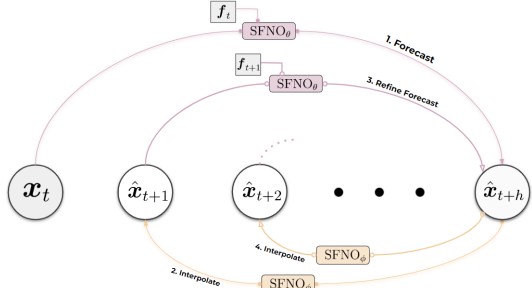

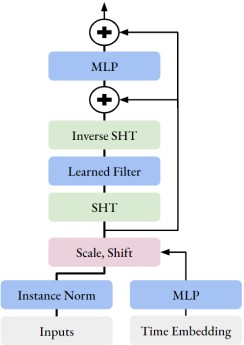

Figure 3: The diagram shows how our proposed approach functions at inference time. Given an initial condition $x_t$ and forcings $f_{t:t+h}$, our method uses the DYffusion framework, integrated with two SFNO backbone networks, to generate predictions for the next $h$ time steps based on an alternation of direct multi-step forecasts and temporal interpolations. To simplify the visualization, we exclude the facts that the interpolator network, $\text{SFNO}_\phi$, is conditioned on $x_t$ and $f_t$ in addition to an estimate of $x_{t+h}$. We also exclude the time-conditioning of both networks. To forecast more time steps beyond $t + h$, our method is applied autoregressively.

Figure 4: Diagram of one of the blocks of the modified SFNO architecture for our proposed method. The full architecture consists of a sequence of 8 such blocks. Our newly introduced time-conditioning modules correspond to the Time Embedding, followed by the MLP on the right, and the scale-shift operation. Our method relies on dropout, which is part of the two-layer MLP on the top. SFNO-based baselines use the same architecture and hyperparameters without the time embedding module.

nature, DYffusion was shown to be faster at sampling time and more memory-efficient than standard diffusion models, while matching or outperforming their accuracy.

### 3.3 Spherical Fourier Neural Operator (SFNO)

The SFNO architecture [8] extends the FNO framework from Li et al. [39] to spherical data and symmetries such as the Earth. FNOs efficiently model long-range interactions in the Fourier space, but because the underlying Fast Fourier Transform is defined on a Euclidean domain, this can lead to modeling artifacts. SFNOs overcome this issue by using the spherical harmonic transform (SHT) [14], a generalization of the Fourier transform, instead. The SFNO model achieves higher long-term stability of autoregressive rollouts than the FNO model, showing stable forecasts of Earth's atmospheric dynamics for up to 1-year-long rollouts at six-hourly time steps. The ACE model from Watt-Meyer et al. [67] is based on the SFNO architecture, modifying some of the hyperparameters and the grid used for the first and last SHT of the SFNO. We use the SFNO configuration from ACE in our experiments.

## 4 Spherical DYffusion

SFNO and ACE are deterministic models that cannot be readily used for uncertainty quantification or ensemble-based climate modeling. DYffusion introduces an efficient diffusion-based approach specifically for forecasting problems but only for Euclidean data. Thus, we propose Spherical DYffusion, a deep generative model for data-driven probabilistic climate simulations that carefully integrates SFNO and DYffusion into an unified framework.

DYffusion requires two neural networks that are used for temporal interpolation and direct multi-step forecasts. In the original framework, these are UNet-like networks. For our approach, we propose to replace them with modified versions of the SFNO architecture, which we denote by $\text{SFNO}_\phi$ and $\text{SFNO}_\theta$, respectively.

**Training.** We follow the original training procedure from DYffusion, complementing it with the use of the input-only forcing variables. That is, for a specified training horizon $h$, these networks are

trained in two stages such that for sequences of prognostic data $\boldsymbol{x}_{t:t+h}$ and forcings $\boldsymbol{f}_{t:t+h}$

$$\text{SFNO}_\phi\left(\boldsymbol{x}_t, \boldsymbol{x}_{t+h}, \boldsymbol{f}_t, i \mid \xi\right) \approx \boldsymbol{x}_{t+i}$$
$$\text{SFNO}_\theta(\text{SFNO}_\phi\left(\boldsymbol{x}_t, \boldsymbol{x}_{t+h}, \boldsymbol{f}_t, j \mid \xi\right), \boldsymbol{f}_{t+j}, j) \approx \boldsymbol{x}_{t+h},$$

where $i \in \{1, \ldots, h-1\}$ and we use $j \in \{0, 1, \ldots, h-1\}$, defining $\text{SFNO}_\phi\left(\boldsymbol{x}_t, \cdot, \cdot, 0 \mid \xi\right) = \boldsymbol{x}_t$. In our experiments, we use $h = 6$. Here, $\xi$ refers to the random variable representing the interpolator network's inference stochasticity. We discuss its implementation further below. The forecaster network, $\text{SFNO}_\theta$, is deterministic. The full training scheme is defined in Algorithm 1.

**Inference.** At inference time, we follow the DYffusion sampling scheme based on cold sampling [2]. Essentially, we start with the initial conditions $\boldsymbol{x}_0$ to generate a first forecast of time step $h$ through a forward pass of the forecaster network, i.e. $\hat{\boldsymbol{x}}_h = \text{SFNO}_\theta(\boldsymbol{x}_0, \boldsymbol{f}_0, 0)$. Given this prediction, we can now use the interpolator network to interpolate $\hat{\boldsymbol{x}}_1 = \text{SFNO}_\phi\left(\boldsymbol{x}_0, \hat{\boldsymbol{x}}_h, \boldsymbol{f}_0, 1 \mid \xi\right)$. In practice, cold sampling applies a correction term to this estimate. The prior forecast of $\boldsymbol{x}_h$ can now be refined with $\hat{\boldsymbol{x}}_h = \text{SFNO}_\theta(\hat{\boldsymbol{x}}_1, \boldsymbol{f}_1, 1)$. The alternation between forecasting and interpolation continues until $\text{SFNO}_\phi$ predicts $\hat{\boldsymbol{x}}_{h-1}$ and the forecaster network performs a last refinement forecast of time step $h$, conditioned on the time $j = h - 1$ and interpolated sample $\hat{\boldsymbol{x}}_{h-1}$. After this final forecast of $\boldsymbol{x}_h$, the process is repeated autoregressively, starting with $\boldsymbol{x}_h$ as the new initial condition. This slightly simplified sampling process is illustrated in Figure 3 and fully described in Algorithm 2. Repeating this sampling process multiple times using the same initial conditions will lead to an ensemble of samples, thanks to the interpolator network being stochastic.

**SFNO time-conditioning.** To use SFNO as described above, it is necessary to implement time-conditioning modules that allow the interpolator and forecaster networks to be conditioned on the time $i$ and $j$, respectively, given that the original SFNO architecture does not support this. We follow the same approach taken by standard diffusion models [13], which consists of transforming the time condition into a vector of sine/cosine Fourier features at 32 frequencies with base period 16, then pass them through a 2-layer MLP to obtain 128-dimensional time encodings that are mapped by a linear layer into the learnable scale and offset parameters. We scale and shift the neural representations of every SFNO block directly following the normalization layer and preceding the application of the SFNO spectral filter, as shown in Figure 4.

**SFNO inference stochasticity.** A stochastic interpolator network, made explicitly through the random variable $\xi$ above, was shown to be a key design choice in the original DYffusion framework. However, to the best of our knowledge, the SFNO model has been only used for deterministic modeling. We overcome this issue through MC dropout [19], i.e. enabling dropout modules [62] at inference time. Following the original SFNO implementation (of training-time-only dropout), we propose to use a dropout module inside the MLP of each SFNO block. In addition, we enable stochastic depth [28]–also known as drop path–at inference time at a rate of $0.1$. Stochastic depth randomly skips a whole SFNO block. When this happens the whole block reduces to the identity function, since only the residual connection is enabled. To the best of our knowledge, this has not been explored before as a source of inference stochasticity.

## 5 Experiments

### 5.1 Dataset and Experimental Setup

To compare our proposed method against ACE [67], we use the same dataset, training and evaluation setup. The dataset consists of 11 distinct 10-year-long simulations from the state-of-the-art global atmospheric model FV3GFS [73], saved every 6 hours. The forcings consist of annually repeating climatological sea surface temperature (1982-2012 average) and incoming solar radiation. Greenhouse gas and aerosol concentrations are kept fixed. The data was regridded conservatively from the cubed-sphere geometry of FV3GFS to a $1°$ Gaussian grid, and filtered with a spherical harmonic transform round-trip to remove artifacts in the high latitudes. We train on 100 years of simulated data from FV3GFS, and evaluate the models on how well they can emulate a distinct 10-year-long validation simulation (i.e. $H = 14600 = 10 \times 365 \times 4$). The 11 simulations form an initial-condition ensemble, where each simulation is independent of the other–after some discarded spinup time–due to the chaoticity of the atmosphere [32]. For more details, see Appendix B.

## 5.2 Baselines

We compare with the following baselines for climate projection.

- **ACE** [67] applied the SFNO architecture to the FV3GFS dataset described above.
- **ACE-STO**: We re-train ACE but use MC dropout, in the same way how it is applied in $\text{SFNO}_\phi$ for our method, to generate stochastic predictions.
- **DYffusion** [57]: We train DYffusion using the original UNet-based architecture as its interpolator and forecaster neural networks.
- **Reference** [73]: physics-based FV3GFS climate model simulations. We use the ten training simulations to create a 10-member reference ensemble that we use to more robustly estimate the 'noise floor' introduced in [67] and to compare the variability of the reference ensemble with sample simulations from our method. Note that this reference is *not* appropriate for weather forecasts given that it is initialized from different initial conditions.

It is worth noting that ACE also compared their results against a physics-based baseline called C48, which corresponds to running FV3GFS at half the original spatial resolution. This makes C48 around $8\times$ less computationally costly to run compared to the reference simulations but was shown to underperform ACE, which our method is shown to outperform in the experiments below.

For ACE, we directly use the pre-trained model from the original paper. ACE was trained on a next-step forecasting objective based on a MSE loss. For ACE-STO, we re-train ACE from scratch with the only difference being that we use a dropout rate of $10\%$ for the MLP in the SFNO architecture. We use the same dropout rate for the interpolator model, $\text{SFNO}_\phi$, in our method. For both DYffusion and our approach, we choose $h = 6$. That is, these models are trained to forecast up to 36 hours into the future. We use the same training and sampling procedures for both, the only difference being the underlying neural architectures.

**Runtime analysis.** In Table 1, we report the computational complexity in terms of the number of neural function evaluations (NFEs) needed to predict $h$ time steps, and the wall clock runtime for simulating one complete validation trajectory of 10 years. For our method, NFEs is not $3h$ because in the first and last iteration we do not need to actually run line 8 and lines 7 & 8 in Algorithm 2, respectively. Our runtime analysis confirms that the computational overhead at inference time for using our method, is less than $3\times$ as much as for a deterministic next-step forecasting model such as SFNO or ACE. This enables our method to provide significant $25\times$ speed-ups and associated energy savings over using the emulated physics-based model, FV3GFS.

Table 1: Computational complexity of the different deep learning methods in terms of: 1) the number of neural function evaluations (NFEs) needed to predict $h$ time steps. and 2) Total inference runtime (simulating 10 years), including the time needed to compute metrics (in hours:minutes). $N$ refers to the number of diffusion steps which usually ranges between 20 to 1000.

| Method | NFE | Runtime |
|---|---|---|
| ACE / SFNO | $h$ | 01:08 |
| Standard diffusion | $Nh$ | N/A |
| Ours | $3(h-1)$ | 02:56 |
| Physics-based FV3GFS | N/A | 78:04 |
| FV3GFS ($2\times$ coarser) | N/A | 45:38 |

All models were trained on A6000 GPUs using distributed training on 2 up to 8 GPUs, ensuring that the effective batch size remains the same (see Figure 8). For a fair inference runtime comparison measuring the wall clock time needed to simulate 10 years (i.e. one full validation rollout), we run all deep-learning baselines on one A100 GPU. We also include the runtime for the physics-based FV3GFS climate model which was run on 96 cores (24 cores for the $2\times$ coarser version) of AMD EPYC 7H12 processors. The deep learning methods are not only much faster, but also much more energy-efficient than FV3GFS.

For illustrative purposes, we also report the complexity of a standard autoregressive diffusion model [25, 35, 51] approach in terms of the number of neural function evaluations (NFEs) needed to predict $h$ time steps, totaling to $Nh$ where $N$ is the number of sampling steps required to reverse the diffusion process. $N$ usually ranges from between 20 to 1000. This makes the use of such an approach less attractive for climate emulation since the resulting inference runtime would not offer as significant speed-ups over the physics-based reference model.

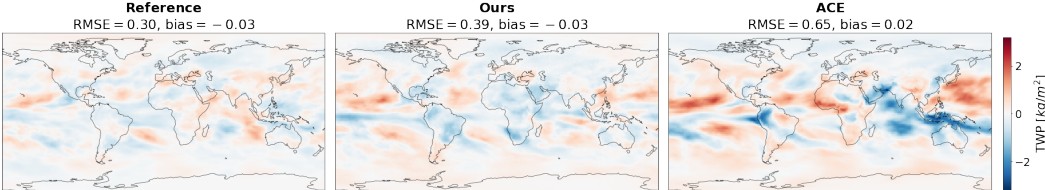

Figure 5: Global maps of the 10-year time-mean biases of a single sample from the reference noise floor simulation, our model, and the ACE baseline for the total water path field. Each subplot reports the global mean RMSE and bias of the respective bias map. Our model reproduces biases of similar location and magnitude to the reference noise floor, suggesting they are mainly due to internal climate variability rather than model bias, while the baseline exhibits larger climate biases.

## 5.3 Climate Biases

**Metrics.** The most crucial quality of an ML-based climate model is its ability to reproduce the climatology of the emulated reference system, i.e. the long-term average ("time-mean") of weather states. The time-mean of the validation simulation is defined as $\frac{1}{H} \sum_{t=1}^{H} x_t$. The time-mean for each model is defined as $\frac{1}{H} \sum_{t=1}^{H} \hat{x}_t$, where $\hat{x}_t$ is the model's prediction for time step $t$. These two quantities are then compared against each other using the bias, i.e. prediction - target, and root mean squared error (RMSE) as metrics of interest for analyzing climate biases. For the probabilistic methods, i.e. ours, DYffusion, and ACE-STO, we generate simulation ensembles by sampling from the model multiple times using the same initial conditions. Unless specified otherwise, all ensemble results are based on $E = 25$ ensemble members. We evaluate the ensemble performance using two metrics: the RMSE of the ensemble-mean prediction ($\frac{1}{EH} \sum_{e=1}^{E} \sum_{t=1}^{H} \hat{x}_{t,e}$) and the RMSE of member-wise time-means ($\frac{1}{H} \sum_{t=1}^{H} \hat{x}_{t,e}$), where $e$ indexes individual ensemble members. For the latter, standard deviations are computed over the member-wise errors. The corresponding "optimal noise floor" for the ML-based emulators is estimated by comparing the validation simulation with the 10-member reference ensemble. All metrics, which are fully defined in Appendix D, are weighted by grid cell area. It is important to acknowledge the potential for improving the estimate of the "noise floor" based on statistical significance testing and improved metrics [21].

**Quantitative analysis.** Our method and all baselines consistently produce stable long-term climate simulations without diverging. In Figure 2, we compare the RMSE of the time-means of the reference, our method, and all baselines.

Our method significantly reduces climate biases compared to baseline methods across most fields, with errors often closer to the reference simulation's noise floor than to the next best baseline. The performance of ACE is notably degraded when made stochastic through MC dropout. Similarly, a direct application of DYffusion fails to accurately reproduce long-term climate statistics. Both these baselines are unable to outperform or even match the scores of the deterministic ACE baseline. Only our proposed careful integration of these two paradigms leads to a skillful climate model emulator: On average, our method's time-mean RMSEs are only $49.36\%$ higher than the noise floor, which is less than half the average RMSE ($110.47\%$) of the next best method, ACE.

Ensemble averaging significantly enhances our method's performance, reducing climate biases by $29.28\%$ on average across all variables. As shown by the light shading in Fig. 2, the ensemble-mean predictions consistently achieve lower time-mean RMSEs compared to single-member predictions (dark shading). This ensemble-based improvement distinguishes our approach from ACE-STO and DYffusion, where ensemble averaging proves less effective, and from ACE, where initial-condition perturbations would be required for ensembling. Additional results for more fields are available in Figure 9 of the Appendix. Our comprehensive evaluation in Table 4 includes ensemble metrics such as the Continuous Ranked Probability Score (CRPS) and spread-skill ratio. The results demonstrate that our method outperforms alternatives in emulating the 10-year time-mean climatology of the reference model for most variables and metrics. However, some challenges remain, particularly in matching the reference ensemble's performance for stratospheric (level 0) variables and in achieving better ensemble scores.

**Qualitative analysis.** In Figure 5 we show the corresponding global maps of the time-mean biases for the total water path (TWP) field. Our model reproduces small biases of remarkably similar location and magnitude to the "perfect-model" reference simulation, with spatial pattern RMSEs of approximately 1% of the global-time-mean TWP. The perfect-model bias is due to unforced random decadal variability in the mean climate of the reference model - each 10-year period has randomly different weather, leading to a slight difference in 10-year time-mean averages across this weather. The reference bias is due to comparing one such decade simulated with the reference model with other simulated decades; its spatial pattern depends strongly on which decade is used for computing the reference model climatology. That our model (trained on 100 years of output) reproduces this pattern suggests that it emulates the long-term (e.g. century-long) time-mean statistics of the reference model even more accurately than a 10-year-mean RMSE can reliably resolve. On the other hand, the baseline ACE model exhibits somewhat larger climate biases, indicative of an actual, albeit small model deficiency that is already evident with a single 10-year estimate of climatology.

In Appendix E.4, we visualize two sample 10-year trajectories simulated by Spherical DYffusion as well as the corresponding validation simulation from FV3GFS. Supplementary videos demonstrate the full temporal evolution of key derived variables: near-surface wind speed[3] and total water path[4]. The emulated fields demonstrate high realism, closely mimicking the patterns and variability observed in actual climate model outputs. This showcases Spherical DYffusion's capability to generate plausible and physically consistent climate scenarios over decadal timescales.

**Climate variability.** Above, we have verified that sampling 10-year-long trajectories from our model produces encouragingly low ensemble mean and member-wise time-mean biases. An important feature of climate is its natural variability on time scales of years, decades, or even centuries even when external forcings (e.g. sunlight or greenhouse gas concentrations) remain unchanged. For instance, multi-decadal periods of rel-

Table 2: Global area-weighted mean of the spread of an ensemble of 10-yr time-mean's for surface pressure, total water path, air temperature, zonal wind, and meridional wind (the last three at the near-surface level). The climate variability of our method is consistent with the reference model.

| Model | $p_s$ | TWP | $T_7$ | $u_7$ | $v_7$ |
|---|---|---|---|---|---|
| Reference | 19.96 | 0.199 | 0.090 | 0.142 | 0.110 |
| Ours | 23.52 | 0.214 | 0.094 | 0.167 | 0.121 |
| DYffusion | 24.75 | 0.223 | 0.082 | 0.169 | 0.127 |
| ACE-STO | 30.32 | 0.256 | 0.135 | 0.192 | 0.131 |

ative drought have stressed many past human civilizations. The present simulations are more constrained than natural climate variability because they employ a repeating cycle of sea-surface temperature and thus do not allow for feedbacks between the atmosphere, ocean, vegetation, and cryosphere. Nevertheless, an important quality of an ML emulator of the global atmosphere suitable for climate studies is that it simulates a similar level of low-frequency climate variability as the reference model.

Here, we verify that our time-mean ensemble passes this challenging test, measured using the intra-ensemble variability of time-mean averages of a few important climate statistics simulated by 25-member ensembles of the emulators vs. the ten reference simulations. We measure this variability by computing the area-weighted average of the standard deviation of time-means across the ensemble dimension. In Table 2 we show that the resulting global mean variability of the ensemble of time-means of our method is within 10-20% of those of the reference simulations for all tabulated variables (and other predicted fields). DYffusion achieves similarly accurate ensemble variability, while ACE-STO In Appendix E.1.2 we show that the corresponding global maps of the time-mean variability reveal similar spatial patterns. That is, our method generates ensemble climate simulations with decadal variability consistent with the underlying climate model.

**100-year-long simulation.** We evaluate the long-term stability of Spherical DYffusion through a 100-year simulation, a critical timescale for many climate modeling applications. Figure 6 demonstrates the model's robustness through time series of key global mean variables from a single (random) simulation, which completed in approximately 26 hours of wall-clock time. The model generates physically consistent temporal patterns in response to annually repeating forcings. Notably, Spherical

---

[3]https://youtu.be/7lHra7gBiBo
[4]https://youtu.be/Hac_xGsJ1qY

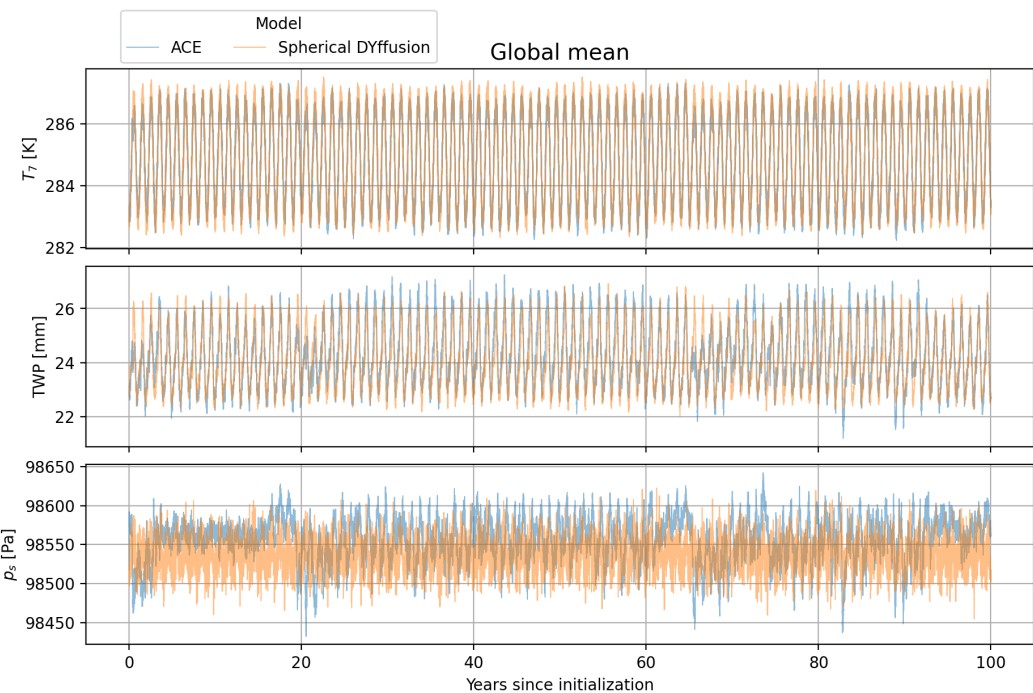

Figure 6: Comparison of 100-year global mean simulations between Spherical DYffusion and ACE. From top to bottom: near-surface air temperature ($T_7$), total water path (TWP), and surface pressure ($p_s$). Both models are driven by identical annually repeating forcings. Spherical DYffusion demonstrates more stable trajectories, particularly evident in the surface pressure predictions, while maintaining physically realistic variability patterns. The consistent behavior across all variables indicates the model's robustness for long-term climate simulations.

DYffusion exhibits improved variability patterns compared to the baseline ACE model, which suffers from unrealistic annual fluctuations (e.g. see surface pressure).

## 6 Conclusion

We introduce Spherical DYffusion, a novel approach that combines efficient diffusion modeling with a spherical-aware neural architecture to probabilistically emulate complex global climate dynamics across decadal to centennial timescales. Our model achieves lower climate biases than relevant deterministic and probabilistic baselines, getting significantly closer to the optimal performance provided by the emulated climate model. For climate model emulation problems, our approach presents a unique solution for balancing generative modeling, computational efficiency, and low climate biases. This opens up the ability to perform fully data-driven ensemble climate simulations.

**Limitations.** To achieve real-world impact, the dataset will need to be expanded so that ML emulators can be evaluated (and trained) on climate change scenarios/simulations. This will require using time-varying climate change forcings such as greenhouse gas and aerosol concentrations. Although our use of the state-of-the-art FV3GFS atmospheric model enables generation of such training data, any emulator will inherently reflect biases present in the base model. Additionally, we only considered emulating the atmosphere, but to achieve a full Earth System Model (ESM) we also need to emulate (or couple to a physics-based model of) other components such as ocean, land, sea-ice, etc. It is important to stress that while our method is more than $25\times$ faster than the reference physics-based climate model, it is still slower than deterministic emulators such as ACE. Though our method characterizes model uncertainty through its generative design, extending it to incorporate initial condition uncertainty—a key component of traditional ensemble physics-based models—could further enhance its capabilities. The method also needs extension to handle output-only variables like precipitation, either through dedicated prediction heads or modifications to the DYffusion framework.

## Acknowledgements

This work was supported in part by the U.S. Army Research Office under Army-ECASE award W911NF-23-1-0231, the U.S. Department Of Energy, Office of Science, IARPA HAYSTAC Program, CDC-RFA-FT-23-0069, DARPA AIE FoundSci, DARPA YFA, NSF Grants #2205093, #2100237,#2146343, and #2134274. S.R.C. acknowledges generous support from a summer internship and subsequent collaboration with the Allen Institute for AI (Ai2), which is primarily funded by the estate of Paul G. Allen. We are grateful to Zihao Zhou, Gideon Dresdner, and Peter Eckmann for their insightful feedback, and to the anonymous reviewers for their constructive comments and valuable suggestions that helped strengthen this work.

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

# Appendix

## A  Broader Impact

The goal of this work is to advance the application of machine learning to climate modeling, specifically for generating fast and cheap ML-based climate simulations. This could significantly democratize climate modeling, improve scientific understanding of the earth system, and enhance decision- and policy-making in a changing climate. However, to realize this goal, the reliability and limitations of such ML models will need to be much better understood.

## B  Dataset

In the subsections below, we elaborate on the dataset and variables that we use, including background information on FV3GFS and how it was configured in order to generate the training and validation data. Any listed data preprocessing steps below are also described in appendix A from [67]. The final training and validation data can be downloaded from Google Cloud Storage following the instructions of the ACE paper at https://zenodo.org/records/10791087. The data are licensed under Creative Commons Attribution 4.0 International.

### B.1  Input, output and forcing variables

Table 3: Input and output variables used in this work. The table was adapted based on Table 1 of [67]. The $k$ subscript refers to a vertical layer index and ranges from 0 to 7 starting at the top of the atmosphere and increasing towards the surface. The two prognostic surface variables, $T_s$ and $p_s$, do not have this additional vertical dimension. Each of their snapshots is a 2D latitude-longitude matrix. The Time column indicates whether a variable represents the value at a particular time step ("Snapshot"), the average across the 6-hour time step ("Mean"), or a quantity that does not depend on time ("Invariant"). "TOA" denotes "Top Of Atmosphere", the climate model's upper boundary.

| Prognostic variables (input and output) | | | | |
|---|---|---|---|---|
| Symbol | Description | Units | Time | Is 3D? |
| $T_k$ | Air temperature | K | Snapshot | Yes |
| $q_k^T$ | Specific total water (vapor + condensates) | kg/kg | Snapshot | Yes |
| $u_k$ | Wind speed in eastward direction | m/s | Snapshot | Yes |
| $v_k$ | Wind speed in northward direction | m/s | Snapshot | Yes |
| $T_s$ | Skin temperature of land or sea-ice | K | Snapshot | No |
| $p_s$ | Atmospheric pressure at surface | Pa | Snapshot | No |
| Forcing variables (input-only) | | | | |
| Symbol | Description | Units | Time | |
| $\mathrm{DSWRF}_{TOA}$ | Downward shortwave radiative flux at TOA | W/m$^2$ | Mean | |
| $T_s$ | Skin temperature of open ocean | K | Snapshot | |
| Additional input-only variables | | | | |
| Symbol | Description | Units | Time | |
| $z_s$ | Surface height of topography | m | Invariant | |
| $f_l$ | Land grid cell fraction | — | Invariant | |
| $f_o$ | Ocean grid cell fraction | — | Snapshot | |
| $f_{si}$ | Sea-ice grid cell fraction | — | Snapshot | |
| Derived, evaluation-only, variables | | | | |
| Symbol | Description | Units | Time | Is 3D? |
| $\mathrm{WS}_k$ | Wind speed | m/s | Snapshot | Yes |
| TWP | Total water path | mm | Snapshot | No |

The complete list of input, output, and forcing variables used in this work is given in Table 3. The only difference to the work from [67] is that we do not consider diagnostic (output only) variables. The forcings consist of annually repeating climatological sea surface temperature (1982-2012 average), $T_s$, and incoming solar radiation, $\text{DSWRF}_{TOA}$. Prescribed sea surface temperatures are simply "overwritten" on the skin temperature predictions of the ML models over all open ocean locations (when rolling out the ML-based simulation). The other forcing or input-only variables are added as an additional channel dimension. Derived variables are computed from the (predicted) prognostic variables as described below.

**Derived variables.** For evaluation, we also consider the derived variable called total water path which is computed as $\text{TWP} = \frac{1}{g} \sum_k q_k^T \, dp_k$, i.e. as a function of surface pressure and the profile of specific total water. Its units are $mm$ (or $kg/m^2$, assuming that water has a density of 1000 $kg/m^3$). The derived wind speed variable for level $k$ is computed based on the simulated meridional and zonal wind variables as $\text{WS}_k = \sqrt{u_k^2 + v_k^2}$. Its units are $m/s$.

## B.2 Background on FV3GFS

Our dataset and physics-based baselines (including our "noise-floor" reference baseline) are based on simulations from a comprehensive global atmospheric model called Finite-Volume on a Cubed-Sphere Global Forecasting System (FV3GFS) [73]. It was developed by the National Oceanic and Atmospheric Administration (NOAA) Geophysical Fluid Dynamics Laboratory (GFDL)[5]. A very similar model version is operationally used by the US National Centers for Environmental Prediction (NCEP) and the US weather forecasting service[6]. Its scalability to horizontal grid spacings as fine as 3 km [12] makes it an excellent candidate for generating training data for future ML-based climate model emulators, including out-of-distribution climate change simulations that may be necessary to train ML emulators on so that they can generalize.

## B.3 FV3GFS configuration for data generation

In the following we summarize the reference data for this study, as also discussed in Section 2.1 of [67]. The training and validation data is generated by running an ensemble of 11 10-year (after discarding a 3-month spinup period) FV3GFS simulations on a C96 cubed-sphere grid (approximately 100 km horizontal grid spacing) with 63 vertical levels. The simulations are an initial-condition ensemble. That is, they are identical except for using different initial atmospheric states. Initial-condition ensembles are a popular tool in climate modeling [32]. Discarding a 3-month spinup period ensures that each simulation is independent of each other due to the chaoticity of the atmosphere[7]. Each simulation is forced by repeating annual cycles of sea-surface temperature and insolation. The temperature, humidity, two wind components at each grid point, and selected vertical fluxes at the surface and top of the atmosphere in each grid column are saved every six hours. For ML training, the temperature, humidity, and two wind components are averaged along FV3GFS's 63 levels to 8 vertical layers, and the data are interpolated to a latitude-longitude grid of $180 \times 360$ dimensions.

# C Implementation details

All methods and baselines are conditioned on the forcings, $\boldsymbol{f}_t$, by simple concatenation of the forcings with the remaining input variables across the channel dimension. We use PyTorch Lightning [16] and Weights & Biases [6] as part of our software stack.

## C.1 Training and inference pseudocode

In Algorithms 1 and 2 we provide the procedures used to train and sample from our proposed method, respectively.

---

[5]https://www.gfdl.noaa.gov/fv3/

[6]https://www.weather.gov/news/fv3

[7]E.g. see Kay et al. [32], who note that: "After initial condition memory is lost, which occurs within weeks in the atmosphere, each ensemble member evolves chaotically, affected by atmospheric circulation fluctuations characteristic of a random, stochastic process (e.g., Lorenz 1963; Deser et al. 2012b)".

---

**Algorithm 1** Spherical DYffusion, Training

---

**Input:** networks $\text{SFNO}_\phi$, $\text{SFNO}_\theta$, norm $\|\cdot\|$, horizon $h = 6$

*Stage 1:* Train interpolator network, $\text{SFNO}_\phi$
    1. Sample $i \sim \texttt{Uniform}(\{1, \ldots, h-1\})$
    2. Sample $\boldsymbol{x}_t, \boldsymbol{x}_{t+i}, \boldsymbol{x}_{t+h} \sim \mathbb{R}^D$, and corresponding forcing $\boldsymbol{f}_t$
    3. Sample network stochasticity (dropout), $\xi$
    4. Optimize $\min_\phi \left\| \text{SFNO}_\phi \left( \boldsymbol{x}_t, \boldsymbol{x}_{t+h}, \boldsymbol{f}_t, i \,|\, \xi \right) - \boldsymbol{x}_{t+i} \right\|^2$

*Stage 2:* Train forecaster network, $\text{SFNO}_\theta$
    1. Freeze $\text{SFNO}_\phi$ and enable its inference stochasticity $\xi$
    2. Sample $j \sim \texttt{Uniform}(\{0, \ldots, h-1\})$ and $\boldsymbol{x}_t, \boldsymbol{x}_{t+h} \sim \mathbb{R}^D$
    3. Retrieve corresponding forcings $\boldsymbol{f}_t, \boldsymbol{f}_{t+j}$
    4. $\hat{\boldsymbol{x}}_{t+j} \leftarrow \text{SFNO}_\phi \left( \boldsymbol{x}_t, \boldsymbol{x}_{t+h}, \boldsymbol{f}_t, j \,|\, \xi \right)$      # with $\hat{\boldsymbol{x}}_{t+j} := \boldsymbol{x}_t$ for $j = 0$
    5. Optimize $\min_\theta \left\| \text{SFNO}_\theta \left( \hat{\boldsymbol{x}}_{t+j}, \boldsymbol{f}_{t+j}, j \right) - \boldsymbol{x}_{t+h} \right\|^2$

---

---

**Algorithm 2** Spherical DYffusion, Inference

---

1: **Input:** Initial conditions $\hat{\boldsymbol{x}}_0 := \boldsymbol{x}_0$, training and inference horizon $h$ and $H = 14600$, forcings $\boldsymbol{f}_{0:H}$
2:  # Autoregressive loop:
3: **for** $t = 0, h, 2 \cdot h, \ldots, (\lceil H/h \rceil - 1) \cdot h$ **do**
4:     # Sampling loop for time steps $t+1, \ldots, t+h$:
5:     **for** $j = 0, 1, \ldots, h-1$ **do**
6:        $\hat{\boldsymbol{x}}_{t+h} \leftarrow \text{SFNO}_\theta \left( \hat{\boldsymbol{x}}_{t+j}, \boldsymbol{f}_{t+j}, j \right)$      # (Refine) forecast
7:        $\tilde{\boldsymbol{x}}_{t+j+1} \leftarrow \text{SFNO}_\phi \left( \hat{\boldsymbol{x}}_t, \hat{\boldsymbol{x}}_{t+h}, \boldsymbol{f}_t, j+1 \,|\, \xi \right)$      # Interpolate
8:        $\hat{\boldsymbol{x}}_{t+j+1} = \tilde{\boldsymbol{x}}_{t+j+1} + \hat{\boldsymbol{x}}_{t+j} - \text{SFNO}_\phi \left( \hat{\boldsymbol{x}}_t, \hat{\boldsymbol{x}}_{t+h}, \boldsymbol{f}_t, j \,|\, \xi' \right)$      # Cold sampling
9:     **end for**
10: **end for**
11: **Return:** $\hat{\boldsymbol{x}}_{1:H}$

---

## C.2   Discussion on the training horizon

The training horizon, $h$, is a critical hyperparameter for both DYffusion and our proposed method. Throughout this study, we use $h = 6$ (corresponding to 36 hours) for both approaches. While we initially explored other horizons, we chose $h = 6$ as it strikes an optimal balance: A smaller horizon (e.g., $h = 3$) reduces the number of sampling steps since the reverse sampling process directly corresponds to physical time steps, potentially degrading performance. Conversely, a larger horizon makes the forecasting task more challenging, as predicting $\boldsymbol{x}_{t+h}$ from $\boldsymbol{x}_t$ becomes increasingly difficult for the forecasting model.

Our choice is further supported by the DYffusion paper, which successfully used $h = 7$ for sea surface temperature forecasting. While we believe that values close to $h = 6$ would likely perform similarly well, comprehensive ablation studies would require re-training two neural networks sequentially, making such experiments computationally expensive to run.

## C.3   Hyperparameters

**Architectural hyperparameters.** To fairly compare against the deterministic SFNO model from [67], we use exactly the same hyperparameters for training the interpolator and forecasting networks for our method, as described in Table 7[8]. For the stochastic version of ACE, ACE-STO, we re-train ACE from scratch with the only difference being that we use a dropout rate of 10% for the MLP in the SFNO architecture. We train the stochastic interpolator model, $\text{SFNO}_\phi$, in our method using the same dropout rate. Both of these stochastic models are run using MC dropout

---

[8]Names correspond to the definition of the SphericalFourierNeuralOperatorNet class found at: https://github.com/ai2cm/modulus/blob/94f62e1ce2083640829ec12d80b00619c40a47f8/ modulus/models/sfno/sfnonet.py#L292. Unless specified otherwise, defaults are used.

Figure 7: Table is directly taken from [67], and reports the SFNO hyperparameters used for ACE as well as the interpolator and forecasting networks of our method.

| Name | Value |
|------|-------|
| embed_dim | 256 |
| filter_type | linear |
| num_layers | 8 |
| operator_type | dhconv |
| scale_factor | 1 |
| spectral_layers | 3 |

Figure 8: Optimization hyperparameters. The effective batch size is calculated as data loader batch size $\times$ number of GPUs $\times$ number of gradient accumulation steps, and is ensured to be the same for all our trained models regardless of the number of GPUs used.

| Name | Value |
|------|-------|
| Optimizer | AdamW |
| Initial learning rate | $4 \times 10^{-4}$ |
| Weight decay | $5 \times 10^{-3}$ |
| Learning rate schedule | Cosine annealing |
| Number of epochs | 60 |
| Effective batch size | 72 |
| Exponential moving average decay rate | 0.9999 |
| Gradient clipping | 0.5 |

(i.e. enabling the dropout layers at inference time). For our interpolator network, we also use a 10% rate for stochastic depth [28], which is also enabled at inference time. This choice was informed by preliminary experiments focused on training a good interpolator network. There, we found the addition of stochastic depth to slightly improve the interpolator's validation CRPS scores (for the interpolated timesteps 1 to 5) and significantly improve the calibration of the interpolation ensemble based on the spread-skill ratio (averaged across variables from around 0.26 to 0.35). We found worse results when using stochastic depth for ACE-STO at inference time.

**Optimization hyperparameters.** We train the interpolator networks for DYffusion and our method on the same relative L2 loss function used for the baseline from [67], and the corresponding forecaster networks on the L1 loss. The models that we train on our own, i.e. the interpolation and forecasting networks of DYffusion and our method are trained with mixed precision. Inference is always run at full precision. For the non-interpolation networks, we perform early stopping based on the best CRPS averaged over a 500-step (125 days) rollout. More optimization-related hyperparameters are discussed in Table 8.

# D Metrics

Unless specified otherwise, all ensemble results are based on $E = 25$ ensemble members. All metrics are area-weighted according to the size of the grid cell, as described below.

## D.1 Preliminaries

Let $\mathbf{X} \in \mathbb{R}^{E \times I \times J}$ denote an ensemble of predictions, and $\mathbf{Y} \in \mathbb{R}^{I \times J}$ the corresponding targets, where $E$ is the number of ensemble members, $I$ is the number of latitudes, and $J$ the number of longitudes in the grid. In the context of this paper, $\mathbf{Y}$ usually corresponds to the validation, reference 10-year time-mean and $\mathbf{X}$ corresponds to an ensemble of 10-year time-means simulated by the reference climate model (excluding the validation time-mean), our proposed method, or any of the baselines.

Let $w(i)$ denote the normalized latitude-dependent area weights at latitude $i$, such that $\frac{1}{I} \sum_i^I w(i) = 1$, which ensure that spatial means are not biased towards the polar regions (see e.g. Rasp et al. [53]).

## D.2 Member-wise Metrics

We report the average, member-wise area-weighted bias, Mean Absolute Error (MAE) and Root Mean Square Error (RMSE), which are defined as follows

$$\text{Bias} = \frac{1}{EIJ} \sum_{e=1}^{E} \sum_{i,j} w(i)(\mathbf{X}_{e,i,j} - \mathbf{Y}_{i,j}) \tag{1}$$

$$\text{MAE} = \frac{1}{EIJ} \sum_{e=1}^{E} w(i)|\mathbf{X}_{e,i,j} - \mathbf{Y}_{i,j}| \tag{2}$$

$$\text{RMSE} = \frac{1}{E} \sum_{e=1}^{E} \sqrt{\frac{1}{IJ} \sum_{i,j} w(i)(\mathbf{X}_{e,i,j} - \mathbf{Y}_{i,j})^2} \tag{3}$$

For the bias, closer to zero is better, for the MAE and RMSE lower is better.

## D.3 Ensemble Metrics

**Ensemble-mean RMSE.** For a skillful ensemble, the magnitude of the average, member-wise RMSE (see above) can be reduced by computing the RMSE on the ensemble-mean prediction, defined as $\bar{\mathbf{X}}_{i,j} = \frac{1}{E} \sum_{e=1}^{E} \boldsymbol{x}_{e,i,j}$, instead.

$$\text{RMSE}_{\text{ens}} = \sqrt{\frac{1}{IJ} \sum_{i,j} w(i)(\bar{\mathbf{X}}_{i,j} - \mathbf{Y}_{i,j})^2} \tag{4}$$

**Spread-Skill Ratio (SSR).** Following Fortin et al. [18], the spread-skill ratio is defined as the ratio between the ensemble spread and the ensemble-mean RMSE. The ensemble spread is defined as the the square root of the ensemble variance

$$\text{Spread} = \sqrt{\frac{1}{IJ} \sum_{i,j} w(i)\text{var}_e(\mathbf{X}_{e,i,j})}, \tag{5}$$

where $\text{var}_e$ computes the variance of the ensemble. Then, we can compute the spread-skill ratio simply as

$$\text{SSR} = \sqrt{\frac{E+1}{E}} \frac{\text{Spread}}{\text{RMSE}_{\text{ens}}}, \tag{6}$$

where $\sqrt{\frac{E+1}{E}}$ is a correction factor which is especially important to include for small ensemble sizes. Note that this factor is omitted in e.g. WeatherBench-2 [53]. The SSR serves as a simple measure of the reliability of the ensemble, where values smaller than 1 indicate underdispersion (i.e. the model is overconfident in its predictions), and larger values overdispersion. That is, closer to 1 is better.

**Continuous Ranked Probability Score (CRPS).** Following Zamo and Naveau [72], we use the unbiased version of the CRPS [44], which is a proper scoring rule:

$$\text{CRPS} = \frac{1}{IJ} \sum_{i,j} w(i) \left[ \frac{1}{E} \sum_{e=1}^{E} |\mathbf{X}_{e,i,j} - \mathbf{Y}_{i,j}| - \frac{1}{2E(E-1)} \sum_{e=1}^{E} \sum_{f=1}^{E} |\mathbf{X}_{e,i,j} - \mathbf{X}_{f,i,j}| \right] \tag{7}$$

where the first term represents the skill, and the second term represents the spread. The biased CRPS averages over the spread with the factor $\frac{1}{2E^2}$, which is biased–especially for small ensemble sizes–compared to using the unbiased version with the factor $\frac{1}{2E(E-1)}$. Note that common Python packages such as `xskillscore` and `properscoring` use the biased version. Lower is better.

**Note on deterministic models.** For deterministic models like ACE without initial condition ensembling, the ensemble size is trivially $E = 1$, causing $\bar{\mathbf{X}}_{i,j}$ to be identical to $\mathbf{X}$. This results in $\text{RMSE}_{\text{ens}}$ reducing to standard RMSE, MAE equaling CRPS, and a zero spread-skill ratio. To accurately

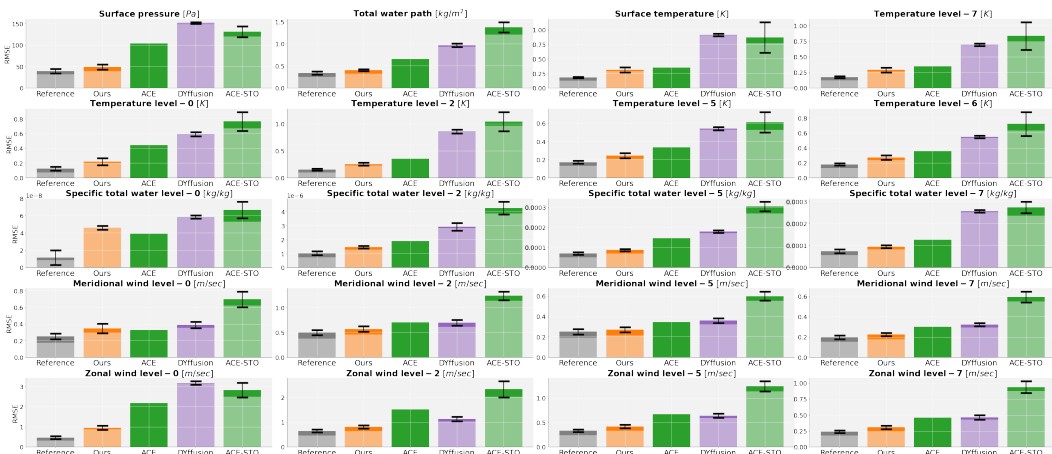

Figure 9: Same as Figure 2 but showing more fields for the RMSE of 10-year time-mean's. Bars (left to right) show 1) the noise floor calculated from the pairwise differences of ten independent 10-year reference model simulations with respect to the validation simulation. In light shade we report the score computed using the mean over the ten reference simulations as "prediction", 2) the member-wise scores of an 25-member ensemble of our method in dark shade, and the corresponding ensemble mean score in light shade, 3) the score of the deterministic ACE baseline, 4) the member-wise scores of an 25-member ensemble of the DYffusion baseline in dark shade, and the corresponding ensemble mean score in light shade. The standard deviation error bar is computed over the set of pairwise (member-wise) time-mean RMSEs for the reference (Ours and DYffusion). ACE does not have a standard deviation since it is a deterministic model. Turning ACE stochastic through MC dropout (ACE-STO) degrades its performance. Our method significantly reduces climate biases over the baseline methods and can be effectively ensembled to reduce its climate biases further, approaching the theoretical lower limit imposed by the noise floor of the reference simulation.

reflect that ensemble metrics are not meaningful for single-member deterministic predictions, we denote these metrics as − for ACE in Table 4. While incorporating initial condition ensembling would enhance ACE's performance on these metrics beyond the naive deterministic baseline, such techniques are orthogonal to the model-based ensembling approaches explored in this work. We leave this extension to future research, noting that initial condition ensembling could potentially improve results for all models in our comparison, including the inherently stochastic ones.

# E    Additional results and figures

## E.1    Climate Biases

We quantitatively analyze the 10-year time mean biases of our model and the baselines in terms of the global mean RMSE in Figure 9. The time-mean prediction is the average over the 14,600 predicted snapshots during the 10 years. Our method significantly reduces climate biases over the baseline methods across most fields. Notably, the errors of our method are often closer to the noise floor of the reference simulation than to the next best baseline. We also show that our method can be effectively ensembled to further reduce climate biases, its ensemble-mean reliably improving time-mean scores across all fields. Interestingly, the stochastic version of ACE, ACE-STO, significantly underperforms the deterministic version. Similarly, the direct application DYffusion fails to match the deterministic ACE baseline, even after ensemble averaging. This shows that MC dropout and DYffusion alone are not the reason for the encouraging performance of our method, but rather the holistic integration of all components, including MC dropout in our SFNO-based interpolator network.

In Table 4, we report a comprehensive evaluation of the (ensemble) of 10-year time-means of each method for a subset of ten representative variables. We report the mean bias error, mean absolute error (MAE), root mean square error (RMSE), ensemble-mean RMSE, spread-skill ratio, and Continuous Ranked Probability Score (CRPS), which are rigorously defined in Appendix D.

Table 4: Comprehensive evaluation of simulated 10-year time-means. Bias, RMSE, and MAE represent average member-wise scores. For Bias (Spread-skill ratio; SSR) closer to 0 (1) is better. For the other metrics, lower is better, with relative changes from the reference shown in parentheses. See Appendix D for mathematical formulations and Table 3 for variable descriptions and units.

| Variable | Metric | Reference | Ours | ACE | ACE-STO | DYffusion |
|---|---|---|---|---|---|---|
| $TWP$ | Bias | 0.004 | **-0.043** | **0.021** | **0.017** | 0.686 |
| | RMSE | 0.336 | **0.404 (+20%)** | 0.653 (+94%) | 1.372 (+308%) | 0.965 (+187%) |
| | $RMSE_{ens}$ | 0.249 | **0.327 (+31%)** | - | 1.206 (+385%) | 0.934 (+276%) |
| | SSR | 1.017 | **0.760** | - | 0.574 | 0.273 |
| | MAE | 0.245 | **0.303 (+24%)** | 0.459 (+88%) | 0.957 (+291%) | 0.768 (+214%) |
| | CRPS | 0.125 | **0.178 (+43%)** | - | 0.639 (+413%) | 0.644 (+417%) |
| $p_s$ | Bias | 0.036 | **4.820** | 45.47 | 34.82 | -126.4 |
| | RMSE | 39.37 | **48.79 (+24%)** | 103.5 (+163%) | 131.1 (+233%) | 151.5 (+285%) |
| | $RMSE_{ens}$ | 31.50 | **39.59 (+26%)** | - | 120.1 (+281%) | 149.0 (+373%) |
| | SSR | 0.847 | **0.766** | - | 0.470 | 0.190 |
| | MAE | 26.26 | **35.60 (+36%)** | 71.69 (+173%) | 93.14 (+255%) | 134.8 (+413%) |
| | CRPS | 14.44 | **21.91 (+52%)** | - | 66.48 (+360%) | 121.6 (+742%) |
| $T_7$ | Bias | 0.011 | **-0.049** | 0.121 | 0.369 | 0.311 |
| | RMSE | 0.172 | **0.290 (+69%)** | 0.349 (+103%) | 0.831 (+383%) | 0.692 (+302%) |
| | $RMSE_{ens}$ | 0.124 | **0.267 (+114%)** | - | 0.734 (+490%) | 0.684 (+450%) |
| | SSR | 1.065 | **0.474** | - | **0.634** | 0.158 |
| | MAE | 0.108 | **0.187 (+73%)** | 0.224 (+108%) | 0.510 (+373%) | 0.408 (+278%) |
| | CRPS | 0.054 | **0.132 (+147%)** | - | 0.343 (+540%) | 0.360 (+573%) |
| $T_5$ | Bias | 0.005 | **-0.068** | 0.079 | 0.173 | 0.377 |
| | RMSE | 0.171 | **0.244 (+42%)** | 0.333 (+94%) | 0.610 (+256%) | 0.540 (+215%) |
| | $RMSE_{ens}$ | 0.132 | **0.211 (+60%)** | - | 0.525 (+299%) | 0.527 (+301%) |
| | SSR | 0.933 | **0.619** | - | **0.657** | 0.228 |
| | MAE | 0.117 | **0.171 (+46%)** | 0.243 (+108%) | 0.451 (+286%) | 0.388 (+232%) |
| | CRPS | 0.060 | **0.110 (+84%)** | - | 0.299 (+402%) | 0.330 (+455%) |
| $T_0$ | Bias | 0.000 | **0.034** | 0.162 | -0.127 | 0.517 |
| | RMSE | 0.124 | **0.220 (+78%)** | 0.444 (+259%) | 0.767 (+520%) | 0.592 (+379%) |
| | $RMSE_{ens}$ | 0.074 | **0.202 (+174%)** | - | 0.674 (+815%) | 0.585 (+695%) |
| | SSR | 1.533 | **0.517** | - | **0.599** | 0.161 |
| | MAE | 0.084 | **0.150 (+78%)** | 0.316 (+277%) | 0.550 (+555%) | 0.526 (+527%) |
| | CRPS | 0.034 | **0.102 (+200%)** | - | 0.348 (+921%) | 0.481 (+1312%) |
| $u_7$ | Bias | 0.012 | **0.038** | -0.170 | **-0.023** | 0.077 |
| | RMSE | 0.240 | **0.307 (+28%)** | 0.456 (+90%) | 0.935 (+289%) | 0.462 (+92%) |
| | $RMSE_{ens}$ | 0.178 | **0.239 (+35%)** | - | 0.874 (+391%) | 0.427 (+140%) |
| | SSR | 1.012 | **0.846** | - | 0.412 | 0.438 |
| | MAE | 0.173 | **0.226 (+31%)** | 0.343 (+98%) | 0.693 (+300%) | 0.339 (+96%) |
| | CRPS | 0.087 | **0.129 (+48%)** | - | 0.519 (+494%) | 0.249 (+185%) |
| $v_7$ | Bias | 0.005 | **0.015** | **0.009** | 0.044 | -0.067 |
| | RMSE | 0.196 | **0.224 (+14.3%)** | 0.299 (+53%) | 0.592 (+202%) | 0.320 (+64%) |
| | $RMSE_{ens}$ | 0.152 | **0.178 (+17.0%)** | - | 0.548 (+260%) | 0.292 (+92%) |
| | SSR | 0.910 | **0.802** | - | 0.439 | 0.471 |
| | MAE | 0.138 | **0.164 (+18.6%)** | 0.224 (+62%) | 0.440 (+218%) | 0.247 (+79%) |
| | CRPS | 0.072 | **0.094 (+30%)** | - | 0.325 (+351%) | 0.179 (+148%) |
| $WS_7$ | Bias | 0.003 | **-0.053** | **-0.017** | -0.080 | **-0.001** |
| | RMSE | 0.243 | **0.303 (+24%)** | 0.437 (+79%) | 0.886 (+264%) | 0.450 (+85%) |
| | $RMSE_{ens}$ | 0.183 | **0.238 (+30%)** | - | 0.823 (+349%) | 0.415 (+126%) |
| | SSR | 0.976 | **0.830** | - | 0.430 | 0.445 |
| | MAE | 0.175 | **0.224 (+28%)** | 0.331 (+89%) | 0.659 (+277%) | 0.334 (+91%) |
| | CRPS | 0.089 | **0.128 (+44%)** | - | 0.488 (+449%) | 0.244 (+175%) |
| $WS_5$ | Bias | 0.022 | **0.058** | -0.104 | **-0.036** | -0.081 |
| | RMSE | 0.324 | **0.398 (+23%)** | 0.626 (+93%) | 1.128 (+248%) | 0.591 (+82%) |
| | $RMSE_{ens}$ | 0.240 | **0.311 (+30%)** | - | 1.030 (+329%) | 0.543 (+126%) |
| | SSR | 1.013 | **0.837** | - | 0.475 | 0.452 |
| | MAE | 0.248 | **0.311 (+25%)** | 0.492 (+98%) | 0.878 (+254%) | 0.456 (+84%) |
| | CRPS | 0.124 | **0.176 (+42%)** | - | 0.636 (+412%) | 0.329 (+165%) |
| $WS_0$ | Bias | 0.151 | **-0.022** | -0.167 | 0.854 | 1.642 |
| | RMSE | 0.450 | **0.944 (+110%)** | 2.163 (+381%) | 2.661 (+491%) | 3.158 (+602%) |
| | $RMSE_{ens}$ | 0.307 | **0.887 (+189%)** | - | 2.349 (+664%) | 3.142 (+922%) |
| | SSR | 1.203 | **0.397** | - | **0.573** | 0.110 |
| | MAE | 0.336 | **0.752 (+124%)** | 1.626 (+384%) | 2.035 (+506%) | 2.044 (+509%) |
| | CRPS | 0.152 | **0.589 (+287%)** | - | 1.354 (+789%) | 1.874 (+1131%) |

### E.1.1    Zonal time-means

In this section, we analyze the absolute magnitudes of the simulated time-means by examining their zonal averages (aggregated over the longitude dimension). We also visualize the standard deviation of the respective ensembles of time- and zonal-means for the reference and stochastic methods. We visualize these in Figures 10 and 11. For several fields, including surface pressure, total water path (not shown), and near-surface temperature (top left subplot in Fig. 10), differences between the simulations are not visually noticeable, except for polar biases in baseline methods. However, discrepancies become pronounced in higher-altitude and wind fields, where our method generally achieves the closest agreement with the reference model. Although near-surface fields are the most relevant for society and decision-making, the clear biases of the baseline method at high-altitude levels might contribute to long-term biases, especially in longer simulations, due to the interactions of atmospheric dynamics across all levels. This observation may partly explain why our method achieves the lowest time-mean biases and RMSEs, as discussed in Appendix E.1.

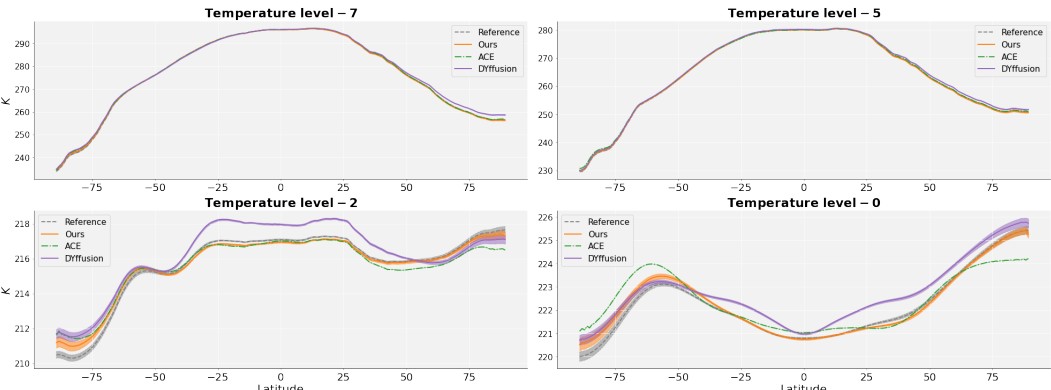

Figure 10: Zonal means of the simulated 10-year time-mean climatologies for a representative subset of four temperature fields. Level 7 represents near-surface conditions, while Level 0 corresponds to the highest altitude. Our method generally provides the closest emulation to the reference data. The most notable biases in the emulations occur at Levels 2 and 0, indicating greater discrepancies at higher altitudes. Emulation challenges are also significant near the poles, including at near-surface levels, particularly for DYffusion.

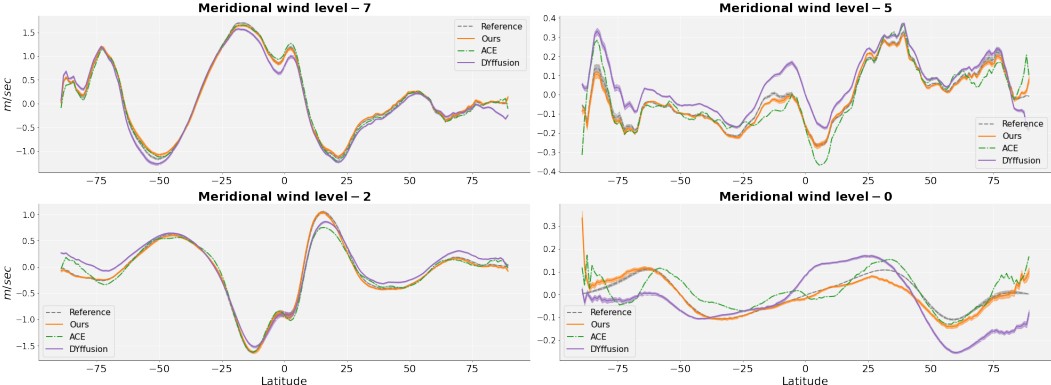

Figure 11: Zonal-means of the simulated 10-year time-mean climatologies for a representative subset of four northward (meridional) wind fields. Our method generally provides the closest emulation to the reference data, except for the level-0 polar latitudes.

### E.1.2 Climate variability

In Fig. 12 we show the global maps corresponding to the global means of Table 2. Our method shows a consistent ensemble variability in terms of the simulated climate that also largely reflects the spatial patterns and magnitudes of the reference ensemble.

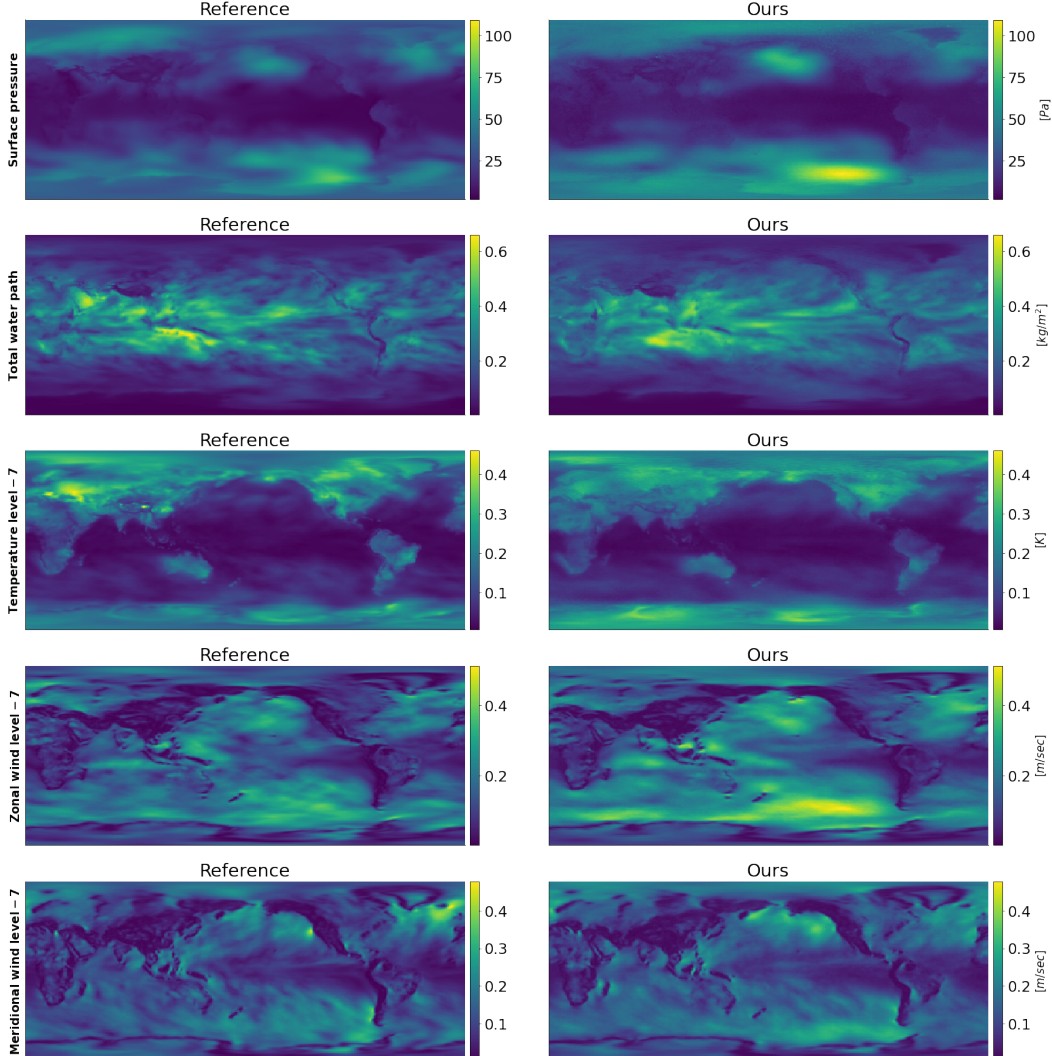

Figure 12: Global maps of the standard deviation of the 10-year time-mean of the reference ensemble and a 25-member ensemble of our method. The climate variability of our method is consistent with the reference model, and largely follows similar spatial patterns with adequate magnitudes. The global mean standard deviation is reported in Table 2.

### E.2 Weather forecasting

While we focus on climate time scales in this work, climate is formed by the statistics of weather, so it is important to verify that our method also generates reasonable forecasts of the weather simulated by the reference model. In Figure 13, we analyze the medium-range forecasting skill of our method and the baselines for lead times up to two weeks. Interestingly, ACE and DYffusion show persistent biases for the surface pressure field that are clearly visible from the first few days of forecasts already but do not seem to reflect on the RMSEs at weather time scales. Such persistent biases, however, may be magnified over longer simulations and could explain why the baselines have problems reproducing accurate long-term climate statistics. In terms of RMSE, the deterministic model ACE generally has a slight edge over our method and DYffusion, especially on lead times of less than a week. After that,

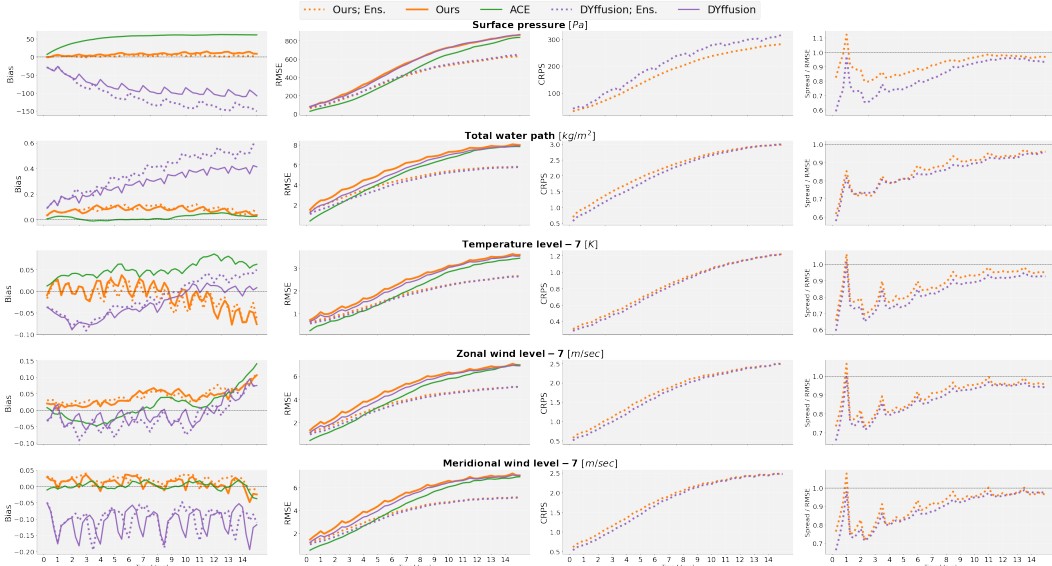

Figure 13: Comparison of medium-range weather forecasting skill between Spherical DYffusion (25-member ensemble and single forecast), DYffusion (25-member ensemble and single forecast), and ACE (single, deterministic forecast). Our method generates competitive probabilistic ensemble weather forecasts, a necessary but not sufficient prerequisite for achieving good climate simulations.

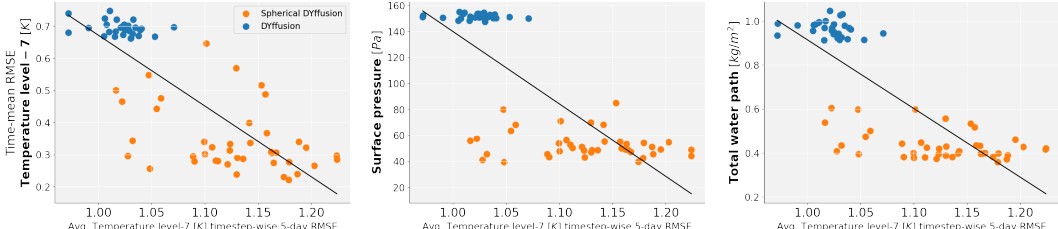

Figure 14: We visualize the performances of DYffusion and Spherical DYffusion (in different marker colors) at multiple checkpoint epochs and for multiple generated samples. We plot the 10-year time-mean RMSE ("climate skill") of three example fields versus the time-step-wise near-surface temperature RMSE averaged out over the first 20 forecasts (5 days; "weather skill"). The 5-day weather forecast performance shows no correlation with the long-term climate biases (indeed, there seems to exist an inverse correlation). This has important implications for practitioners, implying that optimizing for short-term forecasts alone – as is current practice for most ML-based weather forecasting models – may be suboptimal for attaining accurate climate simulations. We have verified that the behavior shown above holds for fields other than near-surface temperature too (not shown).

the ensembles of our method and DYffusion perform best in terms of ensemble-mean RMSE. As expected, the ensemble mean significantly reduces the RMSE compared to using a single sample from our method or DYffusion, especially at longer lead times. The ensemble metrics, CRPS, and spread / RMSE ratio show that our method's and DYffusion's ensemble perform quite similarly, even though they are based on completely different ML architectures. Both ensembles tend to be underdispersed (Spread / RMSE < 1) on short time scales but quickly converge to a well-dispersed ensemble at longer lead times which persists for the whole 10-year climate simulations (not shown).

### E.3 Weather vs. climate performance

In Figure 14, we illustrate that weather performance does not correlate with the climate biases of the same model. We plot the average RMSE over the first 5 days of simulation (here, using the near-surface temperature field) against the 10-year time-mean RMSE of various fields, and do not observe any correlation between the two metrics. We have verified that this observation holds independently of the analyzed field. This is a little-discussed observation that has important implications for ML

practitioners since it implies that optimizing for short-term forecasts alone – as is current practice for most ML-based weather forecasting models – may be suboptimal for attaining accurate climate simulations. Heuristically, optimizing weather skill ensures that a climate model takes a locally accurate path around the climate 'attractor', but it does not guarantee that small but systematic errors may not build up to distort that simulated attractor to have biased time-mean statistics. This observation has been documented for the case of physics-based climate models [17, 54].

### E.4 Qualitative samples

Figures 15, 16, and 17 compare near-surface air temperature, near-surface wind speed. and total water path between the FV3GFS validation simulation, two randomly selected 10-year trajectories generated by Spherical DYffusion, and the trajectory predicted by ACE. For both variables, we show the final ten snapshots of each simulation. The complete temporal evolutions of these simulations for near-surface wind speed and total water path can be viewed at https://youtu.be/7lHra7gBiBo and https://youtu.be/Hac_xGsJ1qY, respectively. The emulated fields demonstrate high realism, closely mimicking the patterns and variability observed in actual climate model outputs. This showcases Spherical DYffusion's capability to generate plausible and physically consistent climate scenarios over decadal timescales.

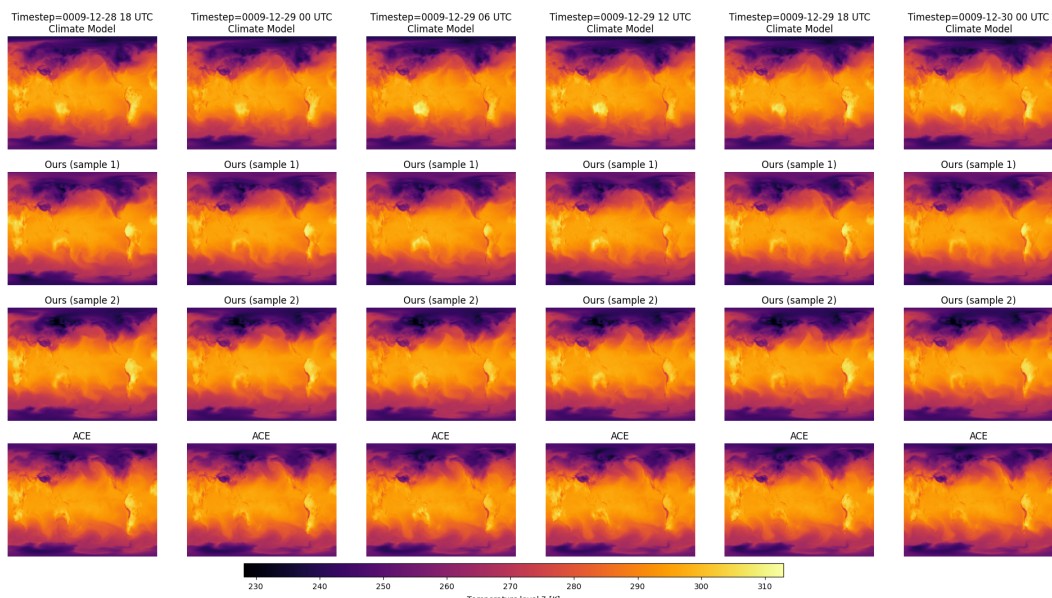

Figure 15: We visualize the final 10 predictions from two random 10-year trajectory samples (i.e. the end of the ninth year) generated by Spherical DYffusion (middle rows) and ACE (bottom row). Here, we show the near-surface air temperature variable, $T_k$ for level $k = 7$. It is important to note that at these extended time scales, simulated trajectories are expected to diverge significantly from one another for any given time step.

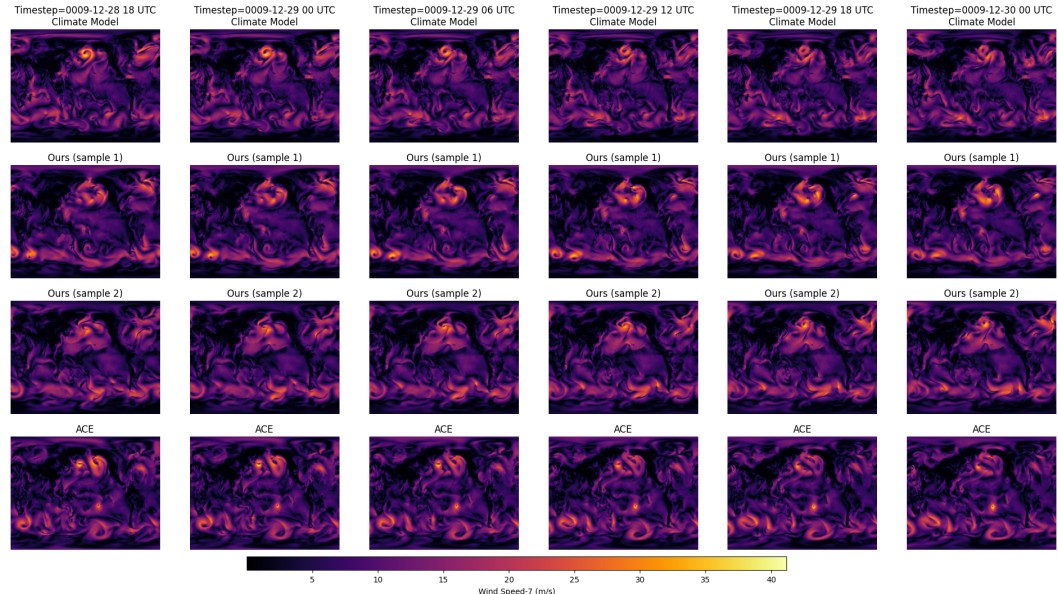

Figure 16: We visualize the final 10 predictions from two random 10-year trajectory samples generated by Spherical DYffusion (middle rows) and ACE (bottom row). Here, we show the derived near-surface wind speed variable, $\text{WS}_k$ for level $k = 7$. It is important to note that at these extended time scales, simulated trajectories are expected to diverge significantly from one another for any given time step. A video visualizing the full 10-year simulations is accessible at https://youtu.be/7lHra7gBiBo.

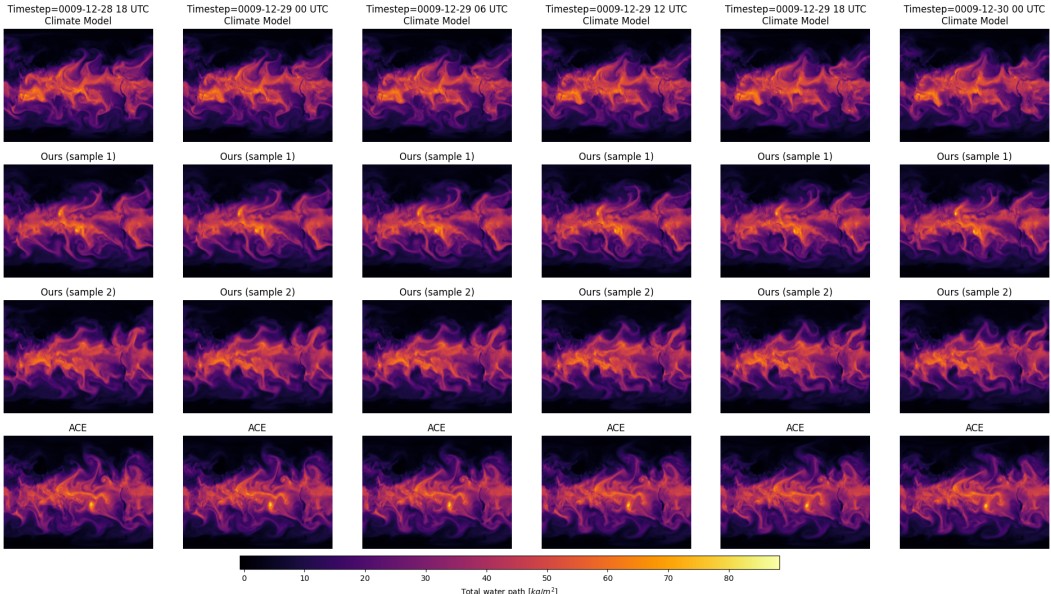

Figure 17: Same as Figure 16 but for the derived total water path variable, TWP. A video visualizing the full 10-year simulations is accessible at https://youtu.be/Hac_xGsJ1qY.

