# OpenReview forum: "Probablistic Emulation of a Global Climate Model with Spherical DYffusion"
_NeurIPS.cc/2024/Conference — NeurIPS 2024 spotlight_

### Official Review · Reviewer_2sHD · 2024-07-01

**Soundness:** 3
**Presentation:** 4
**Contribution:** 4
**Rating:** 8
**Confidence:** 4

**Summary:**

In the paper the authors develop a generative model for global climate simulations. By combining the DYffusion framework for generative modeling with Spherical FNOs that respect the spherical earth geometry the resulting model has stable rollouts for 10-year climate simulations. Experiments show that the resulting model achieves more accurate climate simulations than existing machine learning baselines and the sampled ensemble reproduces meaningful climate variability. Due to the DYffusion framework the generative modeling is achieved without a massive increase in the number of forward passes required to sample a state trajectory.

**Strengths:**

1. ML has a high potential for impact in the development of climate models. As outlines in the paper, ML has seen great success in medium-range weather forecasting and we are starting to see this also translate to climate modeling. This paper significantly pushes this development forward in bringing important advances in terms of both model accuracy and ensemble modeling.
2. The paper is well written and relatively easy to follow, despite containing combinations of multiple complex components (DYfussion, SFNOs, climate modeling). The authors do a good job at making climate modeling concepts accessible to a wider ML audience.
3. The proposed method of combining SFNOs and DYfussion is well-motivated by requirements on stable rollouts + sampling speed + ensemble modeling.
4. The evaluation includes relevant analysis, baselines and metrics.

**Weaknesses:**

1. The stochasticity in the ensemble comes from MC dropout + Stochastic depth in the inference network. There is no clear motivation for this choice.
    1. While MC dropout is an easy way to introduce stochasticity, it is generally known to not represent the true uncertainty very accurately. It is unclear to me if this is a suitable choice for representing the full distribution of possible interpolated states. On the other hand, the SFNO network with MC dropout here sits inside the DYfussion framework. It is now obvious to me how crucial it is for interpolation network to accurately represent the distribution of the atmospheric state for the final ensemble to match the distribution. Perhaps the authors can shed some light on this.
    2. While MC dropout is used in the original DYffusion framework, it is unclear why Stochastic depth was added here. It would be valuable with an ablation to see the importance of MC dropout vs Stochastic depth.
2. The ability of the ensemble to accurately capture climate variability is mainly evaluated in Table 2. In this table I am missing the same spread computed for the (generative) baselines. It is somewhat hard to put the spread of the proposed method into context without comparing to the baselines.

**Questions:**

1. The DYffusion baseline is described as using the original U-net architecture. This is, as accurately represented in the paper, a poor choice that does not take the spherical geometry or any longitudinal periodicity into account. I would expect that this creates issues with artefacts or instability on the "edges" of the flattened earth, and close to the poles. Do you see such problems in practice? How much does these issues explain the poor performance of the DYfussion baseline? I think more discussion about how DYffusion performs empirically could be valuable, as its shortcomings are the main motivation for the use of SFNO in the proposed model.

**Limitations:**

The authors extensively and accurately discuss the limitations of the work.

---

> ### Author Rebuttal · Authors · 2024-08-07
>
> We thank the reviewer for their positive reception of our work and its "important advances". We are especially glad to hear that you think we have done "a good job at making climate modeling concepts accessible to a wider ML audience." We sincerely hope that, similarly to ML-based weather forecasting, we will see a flurry of research on this in the coming years at top ML conferences.
>
> **W1.1:** That is a great point. The main reason for using MC-dropout is its simplicity and the fact that the original DYffusion framework also uses it with promising results. You are right that MC-dropout alone does not lead to a skillful ensemble, but the inclusion of the DYffusion framework is key.  ACE-STO, made stochastic with MC-dropout in the same way as our method's interpolator network, did not show improvement over deterministic ACE (we also verified that stochastic depth does not improve the MC-dropout-only version of ACE-STO).
> A possible explanation is that while MC-dropout is crucial for stochastic sampling, the forecaster's predictions remain deterministic and are used to initialize the new autoregressive rollout iteration.
>
> We will further clarify these in the updated version.
>
> **W1.2:** Thank you for bringing this up. This choice was informed by preliminary experiments focused on training a good interpolator network. There, we found the addition of stochastic depth to slightly improve the interpolator’s validation CRPS scores (for the interpolated timesteps 1 to 5) and significantly improve the calibration of the interpolation ensemble based on the spread-skill ratio (averaged across variables from around 0.26 to 0.35). We will discuss this motivation in our revised draft.
>
> **W.2:** Thank you for the great suggestion. Below, we have added the corresponding spread’s from the baselines. DYffusion reproduces the ensemble spread similarly well as our method, while ACE-STO clearly has troubles reproducing a variability of similar magnitude as the reference.
>
> | Model      | $p_s$ | $TWP$ | $t_7$ | $u_7$ | $v_7$ |
> |------------|---------|---------|---------|---------|---------|
> | Reference  | 19.961  | 0.199   | 0.090   | 0.142   | 0.110   |
> | Ours       | 23.517  | 0.214   | 0.094   | 0.167   | 0.121   |
> | DYffusion  | 24.753  | 0.223   | 0.082   | 0.169   | 0.127   |
> | ACE-STO    | 46.720  | 0.554   | 0.295   | 0.306   | 0.205   |
>
>
> **Q1:** We did not see any obvious visual artefacts at the boundaries of the “image”. However, we did observe biases arise in the DYffusion baseline. For example, Figure 9 shows that DYffusion develops clear biases (left column) for the surface pressure, total water path and meridional wind level-7 variables during the first 90 timesteps that are not (less) present in our method (SFNO-based ACE). Interestingly, these biases are not captured by other weather metrics CRPS, RMSE, spread-skill (right columns). We believe that minimizing such biases is crucial for the long rollouts used to evaluate our climate model emulators, which explains DYffusion's poor performance in terms of climate biases.

---

> > ### Comment · Reviewer_2sHD · 2024-08-09
> >
> > Thanks to the authors for their clarifications. I think both the explanation about why stochastic depth was added and the baseline results in Table 2 will be valuable additions to the paper.
> >
> > As written in my review I think this is an excellent and important paper and I look forward to seeing at NeurIPS!

---

> > > ### Author Response · Authors · 2024-08-10
> > >
> > > Thank you for your thoughtful review and positive feedback. We will include the explanation about stochastic depth and the baseline results from Table 2 in our revised draft. We appreciate your positive assessment and look forward to potentially presenting at NeurIPS. Your input has been valuable in improving our paper.

---

### Official Review · Reviewer_u1ux · 2024-07-07

**Soundness:** 4
**Presentation:** 4
**Contribution:** 3
**Rating:** 8
**Confidence:** 4

**Summary:**

The paper presents a method for approximating a physics-based climate model with a faster data-driven model. To make the model probabilistic the method uses diffusion models (specifically DYffusion). Predictions over longer timescales are generated using autoregressive rollouts. Spherical Fourier Neural Operators (SFNOs) are used to handle the non-euclidean geometry of the earth. Training is performed on a large climate simulation (FV3GFS).

In experiments, the method is compared to a set of alternative approaches. Each method is used to generate an ensemble of climate simulations over 10 years, and the statistics of these runs can be compared to a similar ensemble from the physics-based model. Comparison metrics include the bias and variability of time-averages of atmospheric variables.  In addition to the standard physics-based climate models, experimental comparisons are made to (1) the deterministic ACE model (which uses the SFNO), (2) a probabilistic version of ACE (using MC-dropout), and (3) a DYffusion model with a U-Net architecture instead of SFNO.

**Strengths:**

- The paper is very well written. It is well-organized and the methods used are clearly explained.
- The relation to previous work is clearly explained.
- The experiments compare against strong baselines and support the hypotheses.
- The topic is of significant interest. Physics-based climate models are key to our understanding of climate change, but they are fundamentally limited by the computational costs. There is keen interest in using machine learning to address these challenges by building fast emulators that can be used to generate larger ensembles and help quantify variability and uncertainty. The authors address the key problem by proposing a method that attempts to balance performance with computational complexity.

**Weaknesses:**

- The proposed model is never evaluated using probabilistic performance metrics (such as data likelihood). Diffusion models are attractive for probabilistic modeling because they have been shown to be good at maximizing the training data likelihood (e.g. Song, et al. Maximum Likelihood Training of Score-Based Diffusion Models).  The authors use DYffusion for its speed and say it matches or outperforms the accuracy of standard diffusion models, but is accuracy the right metric for emulating a physical system like this? The experimental results suggest that the model is working well for estimating climate variables, but it would be nice to have a stronger argument that the DYffusion model is a good probabilistic model of this data.

**Questions:**

None.

**Limitations:**

The paper clearly addresses the limitations.

---

> ### Author Rebuttal · Authors · 2024-08-07
>
> We sincerely thank the reviewer for their positive reception of our work, particularly noting its organization, clear relation to previous work, and the practical significance of our contribution to climate modeling and climate change.
>
> **W1:** This is an excellent point. Our paper focuses on standard evaluation assessments for climate modeling. We cannot compute the data likelihood exactly because the diffusion model can be understood as optimizing a variational lower bound. This is complicated further by the DYffusion framework only falling under the umbrella of generalized diffusion models (not a type of Gaussian Diffusion). It is certainly interesting to evaluate the ensemble simulations with probabilistic metrics. In Figure 7, we take a simple approach by evaluating the spread across ensembles of 10-year time-means. Moreover, in Figure 8, we include a comparison of the ensembles of time-means via the Continuous Ranked Probability Score (CRPS), which shows agreement with the ensemble-mean RMSE of the same. However, the CRPS is a pixel-and timestep-wise metric which may not tell the whole story. To our knowledge, more advanced metrics still need to be carefully  developed (for example, https://arxiv.org/abs/2401.14657 presents an interesting approach).

---

### Official Review · Reviewer_JrWU · 2024-07-12

**Soundness:** 4
**Presentation:** 3
**Contribution:** 3
**Rating:** 8
**Confidence:** 5

**Summary:**

In this manuscript, the authors demonstrate a domain-specific, generative approach for a climate model emulator (ACE) trained to reproduce a climate model (FV3GFS) by training on FV3GFS simulation. Their approach is stable and reproduces the climate of the reference climate model with minimal biases. The addition of diffusion-based methodology significantly reduces the biases compared to the original ACE model. The spherical DYffusion approach addresses several important issues: 1.) the long inference and training time for diffusion-based models, 2.) helps introduce correct inductive biases as the atmosphere has spherical geometry, and 3.) allows for climate ensembles.

**Strengths:**

The spherical DYffusion is an important contribution to both the weather/climate domain as it incorporates the spherical nature of the atmosphere and demonstrates the success of the DYffusion approach on a complex, large system. One of the biggest problems with diffusion-based modeling is the long training and inference times (see GenCast where a single forecast takes 8 minutes compared to GraphCast which takes a few seconds). The spherical DYffusion is a good compromise between the benefits of a generative modeling approach and computational time during inference (something important for climate modeling). This approach only increases inference time by a factor of 3 and opens up the possibility for scalable, fast diffusion-based models for weather and climate.

The DYffusion methodology presented in this paper has a large number of applications well outside the weather and climate field.

For climate research, the ability to rapidly run an ensemble of climate simulations is a large selling point for data-driven emulators compared to traditional physics-based, numerical methods.

**Weaknesses:**

Some of the claims in the conclusion are misleading and I suggest toning down the climate change-related conclusions (e.g. "Our method’s climate biases nearly reach a gold standard for successful climate model emulation and thus represent a significant leap towards efficient, data-driven climate simulations that can help society to better understand the Earth and adapt to a changing climate"). Right now this model makes a stationary climate assumption (e.g. boundary conditions and forcings are consistent in both the training data set and during model inference). As mentioned in the limitation section, there is still significant research and evaluation needed before these emulators can start tackling climate change.

The emulator is also trained on climate simulation not reanalysis (e.g. ERA5) which is typically done for data-driven weather models. The FV3GFS itself has biases with respect to the past and present climate which should be noted.

**Questions:**

Does the original ACE model run stability for 10 years without the inclusion of diagnostic variables? In the original ACE paper, there are several claims these variables help with the conservation of energy and moisture, however, the results in this paper suggest that at least in terms of stability, these variables aren't required.

Does the FV3GFS have a good representation of stratospheric dynamics (e.g. QBO and sudden stratospheric warming)? If so how are the stratospheric dynamics in the emulator?

**Limitations:**

The authors are very upfront with the limitations of the work. My only suggestion in the limitation section is to mention this is climate simulation and trained on observation-based products (ERA5) and thus has some biases with respect to the real atmosphere.

---

> ### Author Rebuttal · Authors · 2024-08-07
>
> We thank the reviewer for their positive reception of our work and viewing it as an “important contribution” that has “a large number of applications well outside the weather and climate field”.
>
> **W1:** Sincere thanks for bringing this up. Firstly, we plan to tone it down by rewriting to *“(...) a gold standard for successful climate model emulation under our simplified setting (see limitations)”*. Secondly, we will make clear in our limitations section that a trained emulator should be expected to inherit any biases of the underlying training dataset (here, FV3GFS) with respect to the real atmosphere. We appreciate your insight, which has helped us refine our presentation of results.
>
> **Q1:** That is an interesting question. Based on training our own method as well as DYffusion baseline, our intuition is that yes, stability would not be hurt by removing the diagnostic variables from the training set, but we haven’t actually re-trained ACE this way. Also, from reading the paper it seems to us that the main motivation for including these variables was to be able to evaluate how well ACE conserves energy and moisture rather than helping stability.
>
> **Q2:** This is an excellent question that touches on important aspects of stratospheric dynamics. The top model layer in the FV3GFS-derived dataset encompasses 0-50 hPa, lying entirely in the stratosphere. We note that FV3GFS itself does not have a good representation of the QBO. For sudden stratospheric warming, we are unsure ourselves and would have to dig deeply to evaluate this properly. Generally, stratospheric variables seem particularly challenging to learn due to their slower time scales and partial decoupling from tropospheric weather. Indeed, meridional wind and specific total water at level 0 are amongst the few variables where Spherical DYffusion underperforms ACE in terms of RMSE of the time-means. A more detailed analysis of our model's performance on and potential improvements for stratospheric variables could be an interesting direction for future work.

---

> > ### Comment · Reviewer_JrWU · 2024-08-09
> >
> > Thanks for your response.
> >
> > I am happy with the responses to my questions and the inclusion of the climate change-related limitation. I look forward to future work using ACE and this probabilistic version of ACE, especially for any work related to stratospheric dynamics.
> >
> > I have updated my overall score to 8.

---

> > > ### Author Response · Authors · 2024-08-10
> > >
> > > Thank you for your comprehensive review and follow-up. We are pleased that we could address your questions and concerns. We will carefully incorporate our responses to your questions and concerns into the revised manuscript. We appreciate your updated score and your valuable feedback throughout this process.

---

### Official Review · Reviewer_mrTD · 2024-07-19

**Soundness:** 4
**Presentation:** 3
**Contribution:** 4
**Rating:** 8
**Confidence:** 4

**Summary:**

This paper presents a new method for probabilistic emulator of climate models, based on a type of diffusion model with spherical geometry. The results for climate model emulation are generally quite impressive.

**Strengths:**

1. The paper establishes what appears to be a new state of the art for climate model emulation, significantly improving upon a previous effort (ACE) by a very experienced team of AI/climate scientists.
2. Very clear results, showing large improvements both in reduced bias, and reaosnably variability
3. The method is quite computationally efficient compared to standard diffusion models.

Overall, this problem makes a very significant contribution to the field of AI-based climate modeling, and I strongly support publishing it at NeurIPS.

**Weaknesses:**

1. ACE is stable for up to 100 year time integration. Is this model similarly stable? My default assumption is that this model is not, which indicates it is not necessarily an improvement in practice. (If I'm mistaken, please correct me and the manuscript!)

2. This is not necessarily a weakness of this paper, per se, but I think the "noise floor" approach of ACE for estimating the significance of improvements in terms of reduced bias could be significantly improved upon. You are attempting to estimate a difference in means, so errors and statistical signifiance can be estimated using the central limit theorem, e.g., by taking the average over each year as a (mostly) statistically independent sample.

3. Using CRPS to measure distance between probability distributions, as is done in Figure 8, is a weird choice. It would be better to use a distance measure appropriate to distributions, like the Earth mover's distance.

**Questions:**

1. The authors do not include any images of generate fields. This makes it hard to qualitatively understand how well the model works, e.g., if it looks realistic. It would be informative to show images of fields at the end of a 10 year simulation and compare them to the training dataset. I suspect such comparisons could be very informative, e.g., I would expect that ACE may show more blurry predictions than this new approach. I would encourage the authors to try to find qualitative examples of how their model obtains better results.

2. The broader point about a lack of correlation between weather and climate model accuracy is well-taken, but I would omit the calculation of a correlation coeffient combining both types of dyffusion models. Within each type of dyffusion model, there is a not a clear trend in performance, as is clear from looking at Figure 10.

3. Do you use the "fair" CRPS (which is bias corrected for ensemble size)? This is not a major concern for sufficiently large ensembles, but should be clarified. On a similar note, the spread-skill ratio should not be expected to be 1 for finite ensembles. There is a [correction factor](https://journals.ametsoc.org/view/journals/hydr/15/4/jhm-d-14-0008_1.xml) of sqrt(1+1/m), which I believe would explains away at least the apparent under-dispersion of total water path for the reference simulation.

4. Why do you set h=6? This is worthy of some discussion.

5. Nit: Line 110 references FuXi and GenCast for ensemble weather prediction. I would omit FuXi, given its extremely underdispersed ensemble and poor results, and consider referencing NeuralGCM instead.

**Limitations:**

Yes

---

> ### Author Rebuttal · Authors · 2024-08-07
>
> We thank the reviewer for their positive reception of our work and “strongly” supporting its publication at NeurIPS.
>
> **W1:** That is an excellent point. We did not try running our method for 100 years. While our current simulations cover 10-year segments due to reference data limitations, we recognize the importance of longer-term stability. We are actively working on 100-year inference runs and aim to provide preliminary results during the rebuttal period, with a full analysis to follow in the revised manuscript.
>
> **W2:** Thank you for the suggestion. We are eager to improve the estimate of the “noise floor”. For consistency with ACE, we follow their approach and complement it with the estimation of the “ensemble noise floor”. We will include a discussion of alternative methods for estimating the 'noise floor' in our revised manuscript, acknowledging the potential for improving the estimate using approaches based on the central limit theorem.
>
> **W3:** We appreciate your suggestion. Our choice for the CRPS to measure the quality of the ensembles of time-means was mostly due to its popularity in the time series and, especially,  weather forecasting literature. We agree that better approaches exist (as also discussed with reviewer u1ux) and will note so in our revised Fig. 8 and corresponding text in the Appendix.
>
> **Q1:** Thank you for the great suggestion. We have created videos of two random sample 10-year trajectories by Spherical DYffusion and shared it with the AC (since we are not allowed to directly share links here)*. For the analyzed near-surface wind speed variable (a derived variable from the meridional and zonal wind predictions), we see promisingly realistic outputs compared to the validation climate model simulation.We would be happy to include these visualizations in the final paper.
>
> *We have also included the corresponding snapshots as a set of images in our rebuttal PDF (in case the AC is not able to share the videos with you for any reason). If possible, we would encourage you to look at the videos.
>
> **Q2:** We appreciate this feedback. We will remove the correlation computation from the figure.
>
> **Q3:** Great points. We use the basic formulation of the CRPS as implemented in the python packages properscoring and xskillscore. We will make sure to clearly state this in our updated draft. Similarly, we did not include the correction factor when computing the spread-skill ratio. Note that for the 25-member ensemble, this correction factor is just 1.0198. We will fix this in our revised draft.
>
> **Q4:** This is a great point! The training horizon, $h$, is indeed an important hyperparameter. In our work, we briefly experimented with other horizons (3 and 9) at the initial stage of our project, but then decided to stick with $h=6$ for the following reasons: We believe that it sets a sweet spot between being too small and too large. When it is too small (e.g. 3), it immediately reduces the number of sampling steps for DYffusion and our method, since the reverse sampling process directly corresponds to the time steps $t=0, 1, …, h-1, h$ which can lead to subpar performance. If it is too large, we run the risk that predicting $\mathbf{x}_h$ from early time steps (e.g. based on $\mathbf{x}_0$) is too challenging for the forecasting model. Additionally, the DYffusion paper used a similar horizon of $h=7$ for their sea surface temperature forecasting experiments, which is the data that is most similar to ours. We believe that using $h=9$ or similar values close to $6$ would probably work fine too, but we did not have the compute to run ablations to support this (every new choice of $h$ would require re-training two neural networks sequentially). We will add a discussion of this choice to our revised paper, acknowledging its importance and the rationale behind our selection.
>
> **Q5:** Thank you for the reasonable suggestion, we will adapt our draft to reflect it.

---

> > ### Author Response · Authors · 2024-08-08
> >
> > Dear reviewer mrTD,
> >
> > As promised, we have an **update regarding a 100-year inference rollout** of our method, Spherical DYffusion.
> >
> > Key results:
> > - Our method is stable.
> > - The time series of the global mean for all variables look very promising given the annually repeating forcings that are used. No obvious biases can be observed. We have shared an anonymous link with the AC for a figure that showcases these global means for air temperature at level 7, total water path, and surface pressure, and compares them against ACE.
> > - The global mean time series for our method does not have (or has to a lower degree) the issue observed by ACE of "unrealistic fluctuations on annual timescales". This is clearly visible for surface pressure in the bottom row of the figure shared with the AC.
> >
> > Note that we only ran our method once (taking around 26 hours to complete). If you think it is important, we can work on showing ensemble 100-year simulations in our revised paper.
> >
> > Thank you for your time. Please let us know if you have any other feedback or questions.

---

> > > ### Comment · Reviewer_mrTD · 2024-08-08
> > >
> > > Thanks for your detailed response.
> > >
> > > I am satisfied that the simulations appear realistic, and pleased to hear that the 100 year run was successful. (I don't think it's particularly important for this paper to include ensemble results for the 100 year run, though I do think in the long term this would be valuable for the research community that tries to compare to your work in the future.)
> > >
> > > I have updated my overall score to 8.

---

> > > > ### Author Response · Authors · 2024-08-10
> > > >
> > > > Thank you for your thorough review and for taking the time to consider our additional information. We will include these additional/new details in our revised paper. We appreciate your updated score and your valuable feedback throughout this process.

---

### Author Rebuttal · Authors · 2024-08-07

We sincerely thank all reviewers for their thoughtful, valuable, and encouraging feedback.

We are very encouraged by the consistently positive reception of our work by all reviewers, who value its organization, clear relation to previous work, and the practical significance of our contribution to climate modeling and climate change (u1ux). We appreciate that reviewer mrTD *"strongly"* supports its publication at NeurIPS, while reviewers JrWu and 2HsD see it as an *"important contribution"* and bringing *"important advances"* respectively. We are particularly pleased that reviewer JrWu recognizes its potential for *"a large number of applications well outside the weather and climate field."*

We are especially glad to hear that reviewer 2HsD thinks that we make *"a good job at making climate modeling concepts accessible to a wider ML audience."* We hope this work will catalyze a surge of research in this area of ML-based climate modeling at leading ML conferences, mirroring the recent expansion of ML-based weather forecasting studies.

We have addressed the questions and weaknesses noted by the reviewers in the respective rebuttals.

In response to reviewer mrTD, we attach a PDF that shows qualitative samples from our method as well as the climate model validation simulation. The emulated field demonstrates a high degree of realism, closely mimicking the patterns and variability observed in actual climate model outputs. We have also generated corresponding videos of the full 10-year-long trajectory for the same two random samples of Spherical DYffusion. We shared the anonymous link with the AC (due to the discouraging wording of the rebuttal instructions regarding sharing links) and hope that they can share it with the reviewers.

Thank you again,

The authors of submission 11626

---

### Decision · Program_Chairs · 2024-09-25

**Decision:**

Accept (spotlight)

**Comment:**

The paper proposes an interesting probabilistic emulator for climate models, integrating diffusion models with spherical geometry. The approach, leveraging Spherical Fourier Neural Operators (SFNOs) and DYffusion, offers computational efficiency and accuracy for long-term climate simulations. The method was validated against various baselines, showing superior performance in terms of bias reduction and climate variability.

All reviewers are positive for this paper from the beginning. The authors’ response and additional discussion/results further convinced the reviewers that this work is a solid contribution to weather forecasting as well as climate models. I therefore recommend acceptance of the paper.

The reviewers did mention that some of the claims in the conclusion are misleading, e.g. "Our method’s climate biases nearly reach a gold standard for successful climate model emulation and thus represent a significant leap towards efficient, data-driven climate simulations that can help society to better understand the Earth and adapt to a changing climate". Therefore I encourage the authors to tone down these claims as they promise in the rebuttal.